# The HSP90/R2TP assembly chaperone promotes cell proliferation in the intestinal epithelium

Chloé Maurizy [1,2], Claire Abeza [1,2,9], Bénédicte Lemmers [1,9], Monica Gabola[1], Ciro Longobardi [1], Valérie Pinet[1], Marina Ferrand[1], Conception Paul[1], Julie Bremond[1], Francina Langa [3], François Gerbe[2,4], Philippe Jay [2,4], Céline Verheggen [1,2,8], Nicola Tinari[5], Dominique Helmlinger [6], Rossano Lattanzio [7], Edouard Bertrand [1,2,8,10 ✉], Michael Hahne [1,10 ✉] & Bérengère Pradet-Balade [2,6,10 ✉]

The R2TP chaperone cooperates with HSP90 to integrate newly synthesized proteins into multi-subunit complexes, yet its role in tissue homeostasis is unknown. Here, we generated conditional, inducible knock-out mice for *Rpap3* to inactivate this core component of R2TP in the intestinal epithelium. In adult mice, *Rpap3* invalidation caused destruction of the small intestinal epithelium and death within 10 days. Levels of R2TP substrates decreased, with strong effects on mTOR, ATM and ATR. Proliferative stem cells and progenitors deficient for *Rpap3* failed to import RNA polymerase II into the nucleus and they induced p53, cell cycle arrest and apoptosis. Post-mitotic, differentiated cells did not display these alterations, suggesting that R2TP clients are preferentially built in actively proliferating cells. In addition, high RPAP3 levels in colorectal tumors from patients correlate with bad prognosis. Here, we show that, in the intestine, the R2TP chaperone plays essential roles in normal and tumoral proliferation.

[1] IGMM, Univ Montpellier, CNRS, Montpellier, France. [2] Equipe labélisée Ligue Nationale Contre le Cancer, Paris, France. [3] Centre d'Ingénierie Génétique Murine, Institut Pasteur, Paris, France. [4] IGF, Univ Montpellier, CNRS, INSERM, Montpellier, France. [5] Department of Medical, Oral and Biotechnological Sciences, Center for Advanced Studies and Technology (CAST), 'G. d'Annunzio' University of Chieti–Pescara, Chieti, Italy. [6] CRBM, Univ Montpellier, CNRS, Montpellier, France. [7] Department of Innovative Technologies in Medicine & Dentistry, Center for Advanced Studies and Technology (CAST), 'G. d'Annunzio' University of Chieti–Pescara, Chieti, Italy. [8] IGH, Univ Montpellier, CNRS, Montpellier, France. [9] These authors contributed equally: Claire Abeza, Bénédicte Lemmers. [10] These authors jointly supervised this work: Edouard Bertrand, Michael Hahne, Bérengère Pradet-Balade. ✉email: edouard.bertrand@igh.cnrs.fr; hahne@igmm.cnrs.fr; pradet@crbm.cnrs.fr

The R2TP complex was first discovered in *Saccharomyces cerevisiae* as an HSP90 co-chaperone[1]. HSP90 folds hundreds of substrate proteins (also called "clients") into their native, active state[2]. The chaperone activity of HSP90 is coupled with its ATPase cycle and is regulated by a number of co-factors called co-chaperones. These co-chaperones aid client loading and regulate the HSP90 ATPase cycle[2]. R2TP is an HSP90 co-chaperone but is unusual for two reasons: first, it is composed of four different subunits; second, it is specialized in quaternary protein folding, i.e., it enables the incorporation of clients into multi-subunit complexes (for review[3]). In mammals, R2TP is composed of a heterodimer between PIH1D1 and RPAP3, which associates with a heterohexamer of RUVBL1 and RUVBL2 (Fig. 1a). PIH1D1 and RPAP3 are both involved in substrate recognition[4,5] while RPAP3 also recruits the chaperones HSP90 and HSP70[6,7]. RUVBL1 and RUVBL2 are related AAA+ ATPases that also have chaperone activity[8,9]. Multiple contacts between the PIH1D1:RPAP3 heterodimer and the RUVBL1/2 heterohexamer allow regulation of their ATPase activity[7,10–12]. Importantly, the RPAP3:PIH1D1 heterodimer is specific to R2TP whereas RUVBL1/2 are also part of other complexes such as the chromatin remodelers INO80 and SRCAP[13]. In mammals, R2TP also associates with a set of six prefoldins and prefoldin-like proteins, possibly to help protein folding. Altogether, R2TP associated with prefoldins has been termed the PAQosome, for Particle for Arrangement of Quaternary structure[3].

The first documented R2TP clients were the small nucleolar ribonucleoparticles (snoRNPs), which are required for the maturation of ribosomal RNAs[14,15]. These particles can be grouped into two families, the C/D and H/ACA snoRNPs, each family being characterized by a set of four proteins, which assembles with a variety of small nucleolar RNAs. Later on, other R2TP substrates have been identified, including the U4 and U5 spliceosomal snRNPs[16–18], the nuclear RNA polymerases[19,20], MRE11[21], and the PI3K-like kinases (PIKKs) mTOR, ATR, ATM, DNA-PK, and SMG-1[22]. R2TP clients thus include several multimeric cellular complexes with crucial roles in transcription, ribosome biogenesis, DNA repair and cell growth[3,23], yet the role of R2TP in tissue homeostasis has not been studied. Most of our knowledge about R2TP originates from work in *S.cerevisiae*, mammalian cell lines or in vitro studies with recombinant proteins. In *S.cerevisiae*, R2TP is mostly required under stress conditions[14,24,25]. In *D. melanogaster*, knockdown of *Spag*, *RPAP3* ortholog, compromises early development and germline stem cells, but not somatic organogenesis nor homeostasis[26,27]. The role of HSP90 on tissue development is also poorly documented. Constitutive knockout (KO) murine models showed that one cytosolic paralog of HSP90 was necessary for spermatogenesis while the other one was essential for early development[28,29]. Conditional models to characterize the role of HSP90 and R2TP in specific tissues are still missing.

In this study, we chose to address the role of R2TP in the intestine, the most dynamically self-renewing tissue in adult mammals. The intestine has a well-defined architecture, which is particularly amenable to study cell proliferation and differentiation[30]. Briefly, active stem cells called Crypt Base Columnar (CBC) stem cells reside at the bottom of intestinal crypts, where they divide to either self-renew, or give rise to progenitors (see below). Progenitors migrate through the transient amplifying (TA) compartment to undergo several division rounds and then differentiate. Differentiated cells migrate towards the villus tip, where they ultimately undergo apoptosis and are shed off the tissue[31]. Progenitors can differentiate into enterocytes, responsible for water and nutrient absorption and compose most of the small intestine epithelium. Alternatively, progenitors differentiate into one of the secretory subtypes.

Among these, the goblet cells are the most represented. These cells produce a lubricant protective mucus layer, and their density increases from the duodenum to the colon. In the small intestine, CBC stem cells also generate long-lived Paneth cells (3–6 weeks), which secrete anti-microbial peptides and maintain crypt niche conditions (for review[31]).

In this work, we generated murine models bearing a conditional KO allele of *Rpap3* to study the role of R2TP in intestinal homeostasis. Indeed, RPAP3 is central to R2TP, bridging together HSP90, HSP70, PIH1D1, and RUVBL1/2[7,10–12]. Our work uncovered a crucial role of R2TP in CBC stem cells and progenitors, where it couples the assembly of key cellular machineries with proliferation. In agreement with a link between R2TP and cell proliferation, we found that high RPAP3 expression in human colorectal cancer (CRC) tissues is associated with shorter disease-free survival (DFS) in patients.

## Results

***Rpap3* is an essential gene in mice**. To generate *Rpap3* murine models, we tested different recombinant murine Embryonic Stem cells (ES cells) available from the KOMP consortium (Supplementary Fig. 1a, b). In the *Rpap3*[wtsi] allele, a strong acceptor splice site is introduced before exon 7. The resulting mRNA encodes a Rpap3[wtsi] protein truncated after the first TPR domain, at amino acid 223, followed by a selection cassette and the bacterial β-galactosidase gene (Supplementary Fig. 1a, b). Site-specific recombination at the *Rpap3* locus was confirmed by Southern blot in the H10 clone (Supplementary Fig. 1c). Injection of H10 ES cells into blastocysts generated 3 chimeras with germinal transmission. Intercrossing of *Rpap3*[wtsi/+] animals did not yield any homozygous *Rpap3*[wtsi/wtsi] animal, suggesting that this truncated *Rpap3* allele is homozygous lethal (0/34 pups; Supplementary Fig. 1a, b).

We used the β-galactosidase cassette to characterize *Rpap3* expression. In the small intestine of *Rpap3*[wtsi/+] animals, we detected β-galactosidase activity exclusively at the bottom of the crypts, intercalated between *lacZ* negative cells (Fig. 1b). This could correspond to either CBC stem cells or Paneth cells (see below for a schematic representation of the small intestinal crypts). Pih1d1 interacts with Rpap3, and this interaction is crucial for Pih1d1 stability and R2TP activity[32]. Pih1d1 is detected in CBC stem cells and progenitors but not in Paneth cells nor differentiated cells of the villus (Fig. 1c and below for a schematic representation of the crypts). In the colon, LacZ activity is weak, Pih1d1 is undetectable and both are absent from the stroma (Fig. 1c). Overall, R2TP expression is restricted to proliferative cells.

***Rpap3* is required for small intestine maintenance**. To generate a conditional KO strain, *Rpap3*[wtsi/+] mice were crossed with mice carrying a transgene encoding the Flipase Flp[33]. In the progeny, excision of the Flp-in cassette produced a *Rpap3* allele with exon 7 flanked by *loxP* sites (Supplementary Fig. 1a). *Rpap3*[flox] mRNA encodes a protein identical to wild-type Rpap3. Indeed, there was no bias in transmission nor any obvious phenotype in *Rpap3*[flox/flox] mice. Excision of exon 7 by Cre-recombinase produces a *Rpap3*[Δ7] mRNA with a premature termination codon in exon 8, predicted to be degraded by the nonsense-mediated decay pathway (Supplementary Fig. 1a, b).

To obtain a constitutive deletion of *Rpap3* in the intestinal epithelium, *Rpap3*[flox/flox] mice were crossed with *villin-Cre* (*VilCre*) transgenic mice. The *VilCre* transgene encodes a constitutively active Cre that is specifically expressed in the epithelial cells of the small and large intestine[34]. We were unable to generate homozygous *VilCre; Rpap3*[flox/flox] animals (0/100

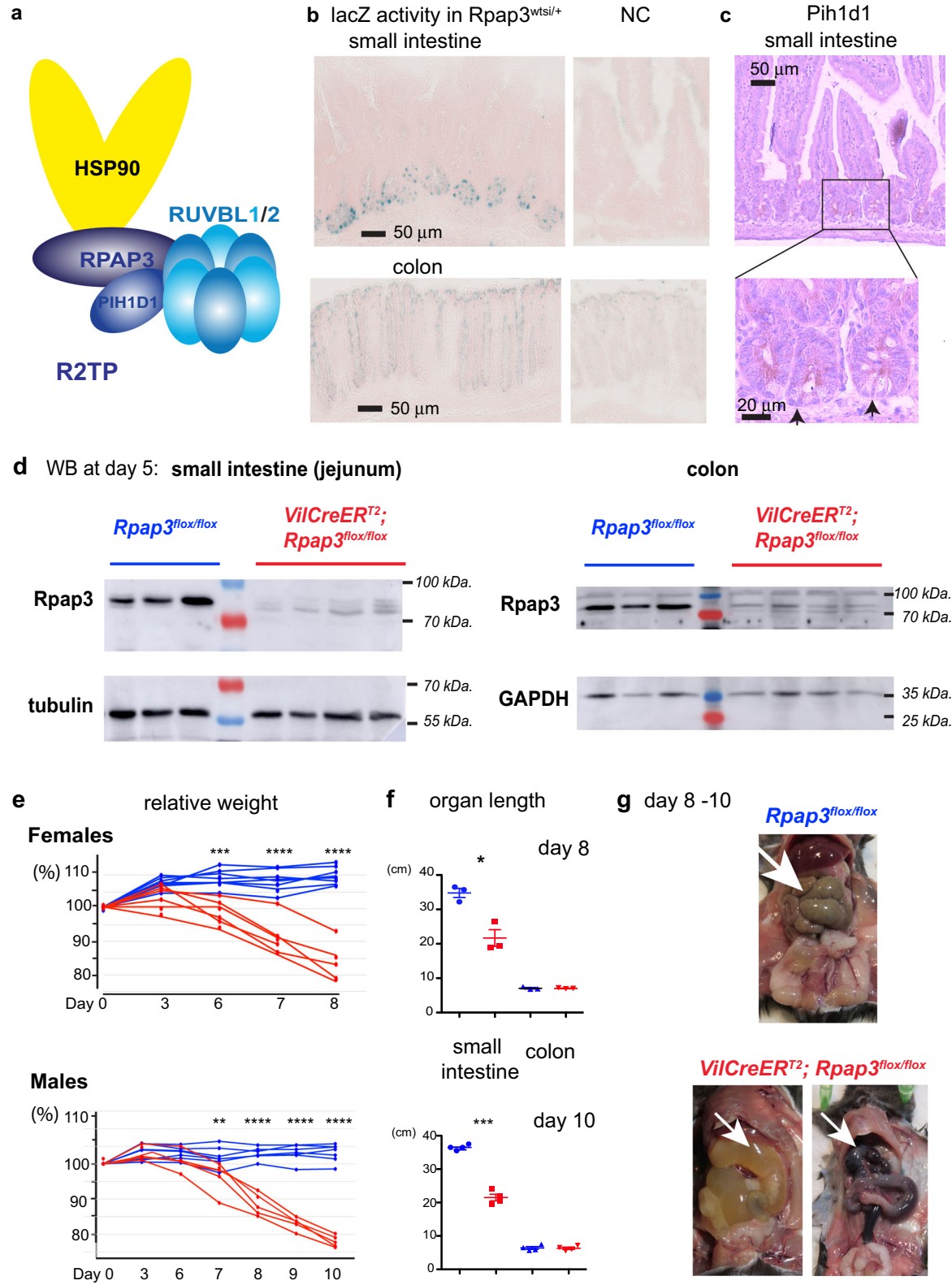

pups; Supplementary Fig. 1a), showing that the absence of *Rpap3* in the intestinal epithelium is lethal. To bypass this problem, we turned to an inducible CreER[T2] under the control of the same promotor *(VilCreER[T2])*. This Cre-recombinase is also specifically expressed in the intestinal epithelium but needs to be activated by tamoxifen[34]. *VilCreER[T2]; Rpap3[flox/flox]* animals were healthy and did not present any obvious phenotype. We then injected

intraperitoneally two doses of tamoxifen separated by an interval of 24 h in 8-week-old mice because the intestine is fully developed at this age. This yielded an efficient recombination, already detectable 1 day after the first injection and only in the small intestine and colon epithelium but not in any of the other organs tested (Supplementary Fig. 1d, e). As a consequence, Rpap3 protein was no longer detected in epithelia from jejunum and

**Fig. 1 *Rpap3* deletion compromises the small intestine and mouse survival. a** Schematic representation of R2TP with its four subunits (RPAP3, PIH1D1, and the RUVBL1/2 heterohexamer). RPAP3 is the core subunit that contacts directly HSP90, PIH1D1 and RUVBL1/2. **b** β galactosidase activity in *Rpap3^wtsi/+* small intestines (top) and colon (bottom), as compared to negative controls (*n* = 2). Scale bars = 50 μm are identical for all images. **c** IHC of Pih1d1 in the small intestine. Counter coloration of DNA with hematoxylin, with magnification (bottom). Top picture: bar represents 50 μm. Inset: arrows point to CBC stem cells, intercalated between Paneth cells with distinctive granules in the cytoplasm; bar is 20 μm. Micrograph is representative of *n* = 3. **d** Depletion of Rpap3 after tamoxifen injection. Western blots were revealed with antibodies against the indicated proteins in extracts of the jejunum (left) and the colonic (right) epitheliums of *RPAP3^flox/flox* controls (blue) or *VilCreER^T2; RPAP3^flox/flox* animals (red), 5 days after the first tamoxifen injection. Each lane was loaded with the lysate obtained from a single animal and were verified for *n* = 12 small intestines and 8 colons, in three independent experiments. Molecular sizes are indicated on the right. **e** Individual weight variations in females (top panel) and males (bottom panel) of tamoxifen treated *VilCreER^T2; RPAP3^flox/flox* animals (red curves, *n* = 6 females and *n* = 5 males) and *RPAP3^flox/flox* controls (blue curves, *n* = 8 females and *n* = 6 males). Individual weights were set at 100% for each animal at day 0, and analyzed by two-way ANOVA (genotype affects weights with *p* < 0.0001 in male and females) and Bonferroni's post hoc multiple comparison tests (**\*\****p* < 0.01; \*\*\**p* < 0.001; \*\*\*\**p* < 0.0001—see Source Data). **f** Length of small intestines and colons from *VilCreER^T2; RPAP3^flox/flox* mice (red points) and controls (blue points) measured at day 8 and day 10 of females (top, *n* = 3) and males (bottom, *n* = 4) were analyzed by two-tailed unpaired *t* test with Welch's correction (females: \**p* = 0.0177, *t* = 4.750, df = 3; males: \*\*\**p* = 0.0001, *t* = 14.14, df = 4). Statistics data represent individual assays with mean ± S.E.M. **g** Representative images of small intestine (white arrowheads) from *VilCreER^T2; RPAP3^flox/flox* animals, which were filled with liquid (left panel) or blood (right panel) from day 8 to 10; a control organ is shown above. Source data are provided as a "Source Data file".

strongly reduced in colon, 5 days after the first tamoxifen injection (Fig. 1d).

*VilCreER^T2; Rpap3^flox/flox* mice showed no obvious phenotype during the first days following tamoxifen injection but suffered from an important weight loss from day 6 onwards, such that most females and males had to be sacrificed from day 8 to 10 (Fig. 1e). The small intestines from these animals were shorter than those of controls and displayed a massive swelling with a transparent liquid or blood (Fig. 1f, g). Histological analysis revealed no alteration in the small intestine of *VilCreER^T2; Rpap3^fox/flox* mice within the first 6 days. At day 7, however, villi epithelial cells presented an altered morphology with surface enterocyte disorganization and focal crowding (Fig. 2a, b red square). At day 8, there was a severe destruction of the small intestinal architecture with villous atrophy and tufts of extruding epithelium (Fig. 2b). In contrast, the intestines of the *VilCreER^T2; Rpap3^flox/+* mice did not show any phenotype (Fig. 2b), suggesting that under normal conditions, one allele of wild-type *Rpap3* is sufficient to maintain tissue homeostasis.

***Rpap3* promotes cell proliferation within the epithelium.** To understand the basis for this rapid degeneration of the intestinal epithelium, we characterized Ki67 expression. Ki67 marks proliferative cells which are, in the intestine, the CBC stem cells and progenitors from the TA compartment (Fig. 3a, b). Immunostaining revealed a loss of Ki67 in the crypts of *VilCreER^T2; Rpap3^flox/flox* animals 7 days after tamoxifen injection (Figs. 2b, 3c). To directly monitor cell cycling, BrdU was injected into the mouse peritoneum 2 h before sacrifice. BrdU is a nucleoside analog incorporated into the DNA during S-phase. BrdU staining was comparable from day 1 to day 6 in the crypts of control and *VilCreER^T2; Rpap3^flox/flox* animals. At day 7 however, there were significantly less BrdU+ cells (Fig. 3d, e). This is coherent with the loss of Ki67 staining and confirms cycle arrest in the crypts and the TA compartment of the mutant animals. At day 8, the atrophic small bowel mucosa showed remnants of crypt glands with some Ki67+ cells (Fig. 2b, black arrowheads). Overall, such epithelium destruction, with remnants of Ki67+ crypt glands, phenocopies TCF4 loss[35]. TCF4 is the transcription factor that integrates Wnt signaling to sustain CBC stemness program[36]. This suggests that R2TP may be required for both CBC and progenitors.

To directly analyze CBC stem cells, we performed immunostaining for Olfm4, a trans-membrane protein commonly used for CBC identification[37] (Fig. 4a, b and Supplementary Fig. 4a). Olfm4 staining was similar between control and *VilCreER^T2; Rpap3^flox/flox* mice from day 4 to day 6 after tamoxifen injection. However, at days 7

and 8, Olfm4 expression was undetectable in most *VilCreER^T2; Rpap3^flox/flox* crypts (Fig. 4a, b and Supplementary Fig. 2a), in agreement with the loss of Ki67+ staining (Figs. 2b, 3c). In the crypts, CBC stem cells intercalate between larger differentiated cells, called Paneth cells. These cells are Ki67− and recognizable with the lysozyme marker (Fig. 4c, d) or by their typical supranuclear eosinophilic (cytoplasmic) granules stained by Hematoxylin/Eosin (HE; Fig. 3c). In marked contrast to CBC stem cells, Paneth cells were detected in a comparable manner in the crypts of control and *VilCreER^T2; Rpap3^flox/flox* mice until day 8 (Figs. 3c, 4d). Altogether, these observations confirmed that the loss of *Rpap3* induces the rapid disappearance of proliferating stem cells and progenitors, but not of differentiated Paneth cells.

Interestingly, we observed cells with round, dense nuclei, resembling apoptotic cells, in the crypts of small intestines deficient for *Rpap3*. We verified this by immunostaining for cleaved caspase 3, a well-established marker of apoptosis. In control mice, few cleaved caspase 3-positive cells were visible at the tip of some villi, where they are normally shed. We detected significantly more cleaved caspase 3+ cells in the crypts of tamoxifen-injected *VilCreER^T2; Rpap3^flox/flox* mice (Fig. 4e). These apoptotic cells were observed in the crypts and TA compartment, but not at the villi tips, as observed in normal epithelia (Fig. 4f). Thus, *Rpap3* deletion eventually induces apoptosis of CBC stem cells and progenitors.

***Rpap3* stabilizes clients of diverse families.** A prominent R2TP client is Rpb1, the largest subunit of RNA polymerase II (RNA PolII)[19,20]. R2TP incorporates neo-translated Rpb1 within RNA PolII in the cytoplasm, after which it is imported into the nucleus[19]. As expected, immunohistochemistry (IHC) for Rpb1 displayed a nuclear staining in control intestines (Fig. 5a). In *VilCreER^T2; Rpap3^flox/flox* animals, 6 days after the first tamoxifen injection, when the epithelial architecture was still preserved, Rpb1 accumulated in the cytoplasm of CBCs and TA progenitors, but remained nuclear in differentiated epithelial cells, including Paneth cells (Fig. 5a). This suggested that Rpap3 is necessary to assemble RNA PolII in the proliferative compartment (Fig. 5b).

Misfolded HSP90 clients are usually degraded[38], so we determined the expression levels of R2TP clients in *VilCreER^T2; Rpap3^flox/flox* animals at day 6. R2TP was initially characterized as an assembly factor for the box C/D snoRNPs, and in particular it chaperones NOP58, one of the four core proteins in these particles[15,39]. Western blot analysis of crypt cell lysates showed a near twofold reduction of NOP58 levels in extracts from *VilCreER^T2; Rpap3^flox/flox* animals, as compared to controls (Fig. 5c). R2TP stabilizes other non-coding RNPs in HeLa cells,

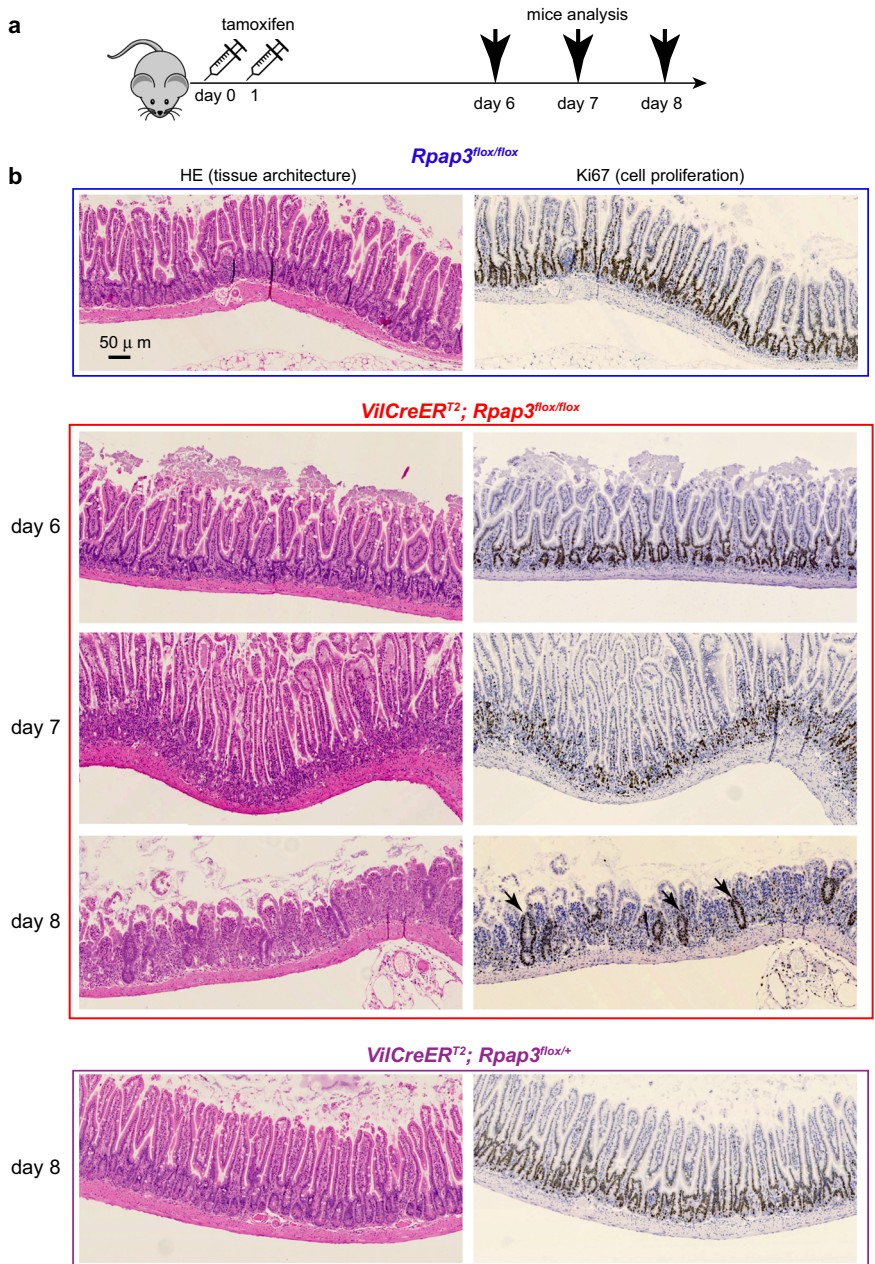

**Fig. 2 Rpap3 is necessary to small intestine integrity. a** Schematic representation of the experimental setting: 8-week-old mice of the indicated genotype received two sequential injections of tamoxifen 24 h apart, and were analyzed 6 to 8 days after the first injection. **b** Pictures of jejunum tissue sections stained with HE (left) or by IHC with anti-Ki67 antibodies (right panels—Ki67 signal is brown) at different days after the first tamoxifen injection. Black arrowheads indicate remnants of crypt glands with Ki67+ cells observed at day 8. Scale bar is identical for all panels, and is 50 μm as shown in control HE panel. Each panel is representative of 8 to 12 animals from three independent experiments.

including the U5 snRNP, a splicing ribonucleoparticle that includes the proteins PRPF8 and EFTUD2[17,18]. Expression of these proteins was reduced by nearly twofold in KO animals, showing their dependency on R2TP in the small intestine (Fig. 5c). HSP90 and R2TP are thought to stabilize clients only before their assembly, and the extreme stability of snoRNPs and snRNPs may explain this moderate but consistent decrease observed here for their components[40].

In mammals, PIKKs consist of six structurally related proteins (ATR, ATM, DNA-PK, mTOR, TRRAP, and SMG1). PIKKs are stabilized by the trimeric co-chaperone TTT[32,41–43] and in murine fibroblasts, TTT recruits R2TP to assemble PIKKs with their partners[22]. By Western blot, we observed again a strong diminution of ATR and ATM, the primary sensors of DNA damage, as well as of mTOR, which activates translation and cell proliferation in response to nutriment availability (Fig. 5d). These results show that R2TP participates in the stabilization of ATR, ATM, and mTOR. Yet, staining for γH2AX, a marker of DNA damage, was similar in control and KO tissues (Supplementary Fig. 3a). In contrast to the other PIKKs, TRRAP level did not vary, either because R2TP does not chaperone TRRAP or because it is a very stable protein.

Interestingly, in control tissues, these R2TP substrates are concentrated in the crypts rather than in the villi. This was observed for the U3 box C/D snoRNP (Supplementary Fig. 3b) and components of the U5 snRNP, as well as ATR

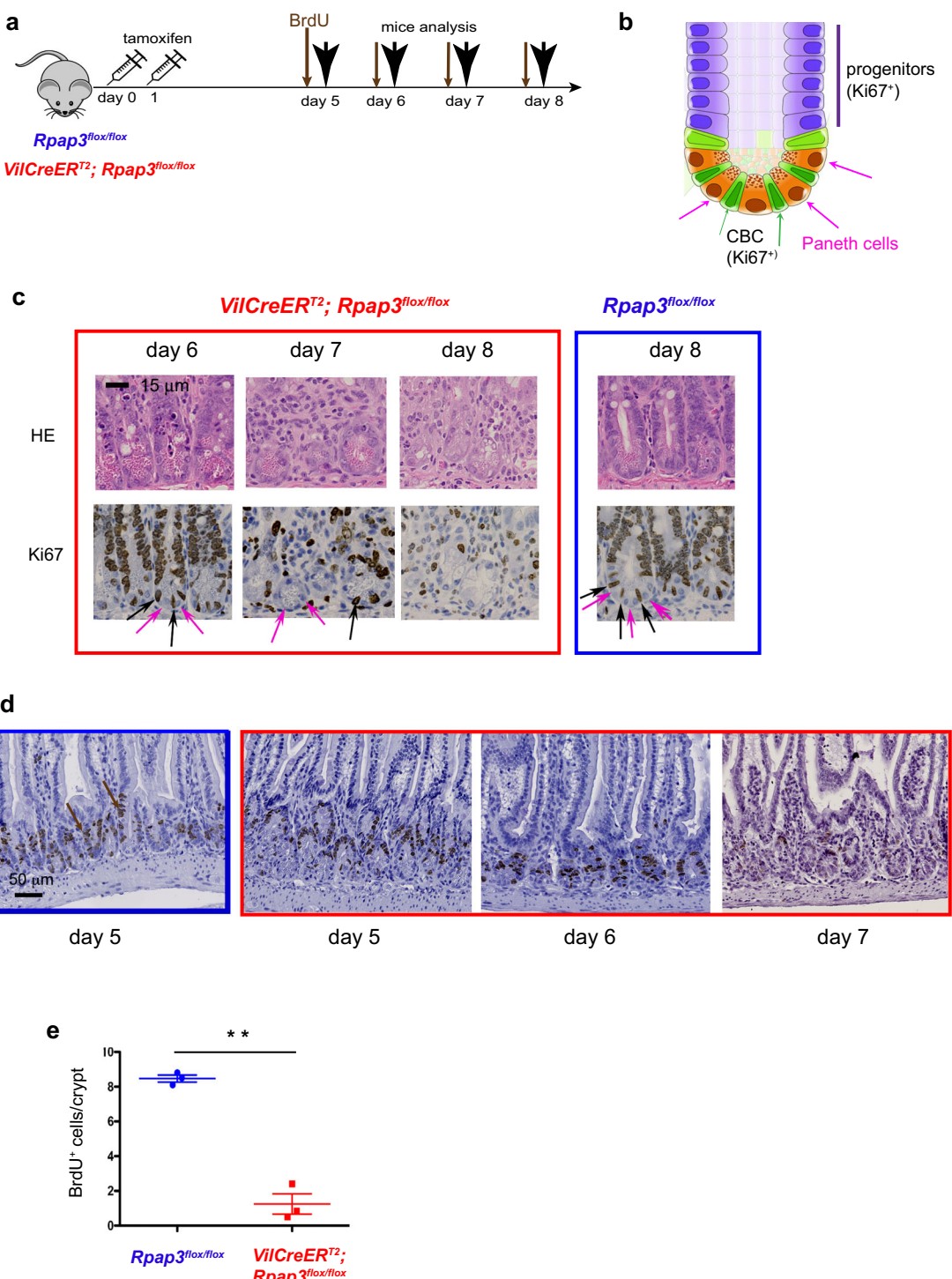

(Supplementary Fig. 3c). Altogether, these data suggest that R2TP assembles cellular machineries mainly in the CBCs and progenitors.

**R2TP *KO* triggers p53-dependent and independent apoptosis.** Cell stress, including DNA damage and ribosome biogenesis defects, stabilize p53, which in turn arrests cell cycle and induces apoptosis. We observed a strong increase of p53 levels at day 6 in the crypts and TA compartment of KO mice (Fig. 6a, Supplementary Fig. 4a). Mutually exclusive staining for p53 or lysozyme confirmed that the induction of p53 occurs in CBCs and

progenitors but not in Paneth cells (Fig. 6b and Supplementary Fig. 4b).

To address the role of p53 in epithelium degeneration following the loss of R2TP activity, we generated double KO *VilCreER^{T2}; Rpap3^{flox/flox}; Trp53^{fox/flox}* mice. We verified that adding floxed alleles did not alter the kinetics of *Rpap3* recombination, which was already detected 16 h after tamoxifen injection (Supplementary Fig. 4c). Accordingly, the phenotype of *Rpap3* deletion was unaffected by hemizygous deletion of *Trp53* as *VilCreER^{T2}; Rpap3^{flox/flox}; Trp53 ^{flox/+}* mice phenocopied *VilCreER^{T2}; Rpap3^{flox/flox}; Trp53^{+/+}* mice (hereafter referred to as "*Rpap3* KO"). As reported before, invalidation of *Trp53* alone did

**Fig. 3 Rpap3 is required for proliferation in the small intestine. a** Schematic representation of the experimental setting: 8-week-old mice of the indicated genotype received two sequential injections of tamoxifen 24 h apart, and were analyzed 5 to 8 days after the first injection. 2 h before each sacrifice, BrdU was injected intraperitoneally to detect cells in S-phase (thin arrows). **b** Schematic representation of a crypt from the small intestine, with CBC stem cells (in green) sandwiched between Paneth cells (in brown) and progenitors forming the TA on top (purple). **c** Representative pictures of jejunum tissue sections stained with HE (top) or by IHC with anti-Ki67 antibody (bottom, Ki67 signal is brown) at indicated days after the first tamoxifen injection. Pink arrows point towards Paneth cells and black arrows towards CBC stem cells. Scale bar is shown in control HE panel. Each panel is representative of 8 to 12 animals analyzed in three independent experiments. Scale bar is identical for all panels and is 15 μm. **d** Representative pictures of jejunum taken from control (boxed in blue) and *VilCreER^T2*; *Rpap3^flox/flox* animals (boxed in red), stained by IHC with anti-BrdU antibodies (brown arrows). *n* = 3–6 animals/ time point from two independent experiments. Scale bar is shown in control panel and is 50 μm. **e** Graph shows mean number of BrdU⁺ cells/crypt in controls and *VilCreER^T2*; *Rpap3^flox/flox* animals at day 7. Each point represents the average number of BrdU⁺ cells calculated in $n > 35$ crypts from two different zones per animal ($n = 3$). Mean values with S.E.M are indicated for each experimental group. Unpaired two-tailed *t* test with Welch's corrections ($t = 11.66$, df = 2) indicates significant difference between control and *VilCreER^T2*; *Rpap3^flox/flox* animals ($p = 0.0073$; $n = 3$). Source data are provided as a "Source Data file".

not cause any noticeable phenotype[44,45] (Fig. 6c and Supplementary Fig. 6d). However, double KO mice showed a transient rescue. At day 6, the intestinal epithelium from *VilCreER^T2*; *Rpap3^flox/flox*; *Trp53^fox/flox* mice appeared normal, with regular Ki67 crypt staining, unlike that of *Rpap3* single KO mice (Fig. 6c). At day 7, the epithelium from double KO mice started shrinking, with partial loss of Ki67 staining, while that of single Rpap3 KO mice were strongly disorganized (Fig. 6c). Finally, at day 8, the epithelium from both single *Rpap3* and double KO showed similar strong villus blunting (Fig. 6c). Staining for p53 and Rpb1 confirmed the near total penetrance of *Trp53* and *Rpap3* deletion (Supplementary Fig. 4d and Fig. 6d). We detected cleaved caspase 3 and typical round apoptotic cells in both the double *Trp53*; *Rpap3* KO and the single *Rpap3* KO crypts, but not in control nor p53 KO crypts (Fig. 6e). Altogether, these results show that inactivation of *Rpap3* induces p53 expression, cell cycle arrest and apoptosis. Still, sustained defective R2TP activity results in epithelium degeneration even in absence of p53.

**Rpap3 functions in the colon as in the small intestine.** Depletion of the Rpap3 protein was effective in both the small intestine and the colon (Fig. 1d). Yet, colons were not shorter in *Rpap3* KO mice (Fig. 1f), nor was their architecture modified until day 8, as observed by staining with Periodic Acid Schiff (PAS), Ki67, BrdU, and CD44v6 labeling of colonic stem cells[46] (Fig. 7a and Supplementary Fig. 5a, b). We analyzed R2TP chaperone activity in the colon by performing Western blots. This revealed a small diminution of R2TP substrates in the colon, mostly visible for mTOR and ATM. This diminution was less pronounced than that in the small intestine (Fig. 7b and Supplementary Fig. 5c). It was not accompanied by any detectable accumulation of Rpb1 in the cytoplasm (Supplementary Fig. 5d), nonetheless p53 was induced at the bottom of the crypt, as well as apoptosis (Fig. 7a, c and Supplementary Fig. 5e). This suggests that the small defects occurring in the colon are sufficient to trigger p53 activation, but not to induce a visible phenotype at the tissue level.

To analyze the role of R2TP in the differentiation of the small intestine and the colonic crypts, we simultaneously generated organoids from both tissues, using mice treated with tamoxifen to delete *Rpap3*, as previously described[47]. Crypts prepared from control small intestines grew to form organoids that budded after 72 h in culture. In contrast, organoids prepared from KO mice degenerated from 48 h on, with no observable budding, confirming the essential role of *Rpap3* for the survival and differentiation of the small intestinal crypts (Fig. 7d). In parallel, control colonic organoids started differentiating at 72 h and budded after 96 h in culture. Remarkably, colonic organoids from *Rpap3* KO mice also died without reaching the budding stage, at 72 h (Fig. 7d). Thus, R2TP is necessary for the survival and differentiation of both, small intestinal and colonic organoids.

**Rpap3 phenotypes mirrors cellular turnover.** To reassess the role of *Rpap3* in the colon, we generated *Lgr5-EGFP-IRES-CreER^T2*; *Rpap3^flox/flox* animals (Supplementary Fig. 6a). In these mice, GFP and the tamoxifen-inducible Cre are both under the control of the CBC⁻specific *Lgr5* promoter. This construct has a mosaic expression restricted to a few GFP⁺ crypts[48], allowing the animals to survive and the comparison of Rpap3-deficient and -expressing crypts in the same intestines. Indeed, tamoxifen treated *Lgr5-EGFP-IRES-CreER^T2*; *Rpap3^flox/flox* mice did not show any visible physiological alteration and could survive for 3 weeks at least. Co-staining for GFP and Rpb1 showed that, at day 7, in the small intestine, Rpb1 accumulated in the cytoplasm of CBC stem cells and progenitors from GFP⁺ crypts, but not in GFP⁻ crypts, nor in non-recombinant GFP⁺ crypts from *Lgr5-EGFP-IRES-CreER^T2*; *Rpap3^+/+* control mice (100% of GFP⁺ crypts had cytoplasmic Rpb1; Fig. 8b and Supplementary Fig. 6a). At day 10, GFP⁺ crypts had been eliminated from the small intestine of *Lgr5-EGFP-IRES-CreER^T2*; *Rpap3^flox/flox* mice, as expected from ex vivo organoid experiments (Fig. 7d). In the colon, Rpb1 was nuclear in GFP⁻ and GFP⁺ colonic crypts from *Lgr5-EGFP-IRES-CreER^T2*; *Rpap3^flox/flox* mice at day 7 (Supplementary Fig. 6a, b). At day 10–12, however, Rpb1 accumulated in the cytoplasm of CBC stem cells and progenitors of GFP⁺ crypts, but not in GFP⁻ and GFP⁺ crypts from control animals (Fig. 8b and Supplementary Fig. 6b). Thus, Rpap3 is necessary for Rpb1 biogenesis in both the small intestine and the colon, yet the effect on Rpb1 appears later in the colon than in the small intestine.

To understand the reason for this delay, we compared the cellular turnover between these tissues. For this, we injected the nucleoside analog BrdU as a cell tracer into wild-type mice and sacrificed them subsequently at different time points (Fig. 8c). Two hours after injection, BrdU was incorporated exclusively by CBC stem cells and progenitors in both the small intestine and the colon (Fig. 8c). After one day, BrdU⁺ cells were detectable above the TA compartment in the small intestine, while some remained at the bottom of the crypt. BrdU⁺ cells then migrated to the tip of the villus at day 3, and were eliminated at day 4, in accordance with the previously described kinetics[49]. In contrast, in the colon, migrating BrdU⁺ cells did not reach the crypt tip before day 4 and were still detectable at day 5 (Fig. 8c). This illustrates a slower cellular turnover in the colon than in the small intestine, which correlates with a slower accumulation of Rpb1 in the cytoplasm of colonic cells, smaller effects on client levels and milder phenotypes at the tissular level (Fig. 7 and Supplementary Fig. 5).

**RPAP3 expression correlates with colorectal cancer prognosis.** Our results so far suggest an intimate link between R2TP activity and cell proliferation in the intestine. To test if R2TP is also involved in pathogenic proliferation, we took advantage of the CODREAD dataset (available at https://xenabrowser.net/).

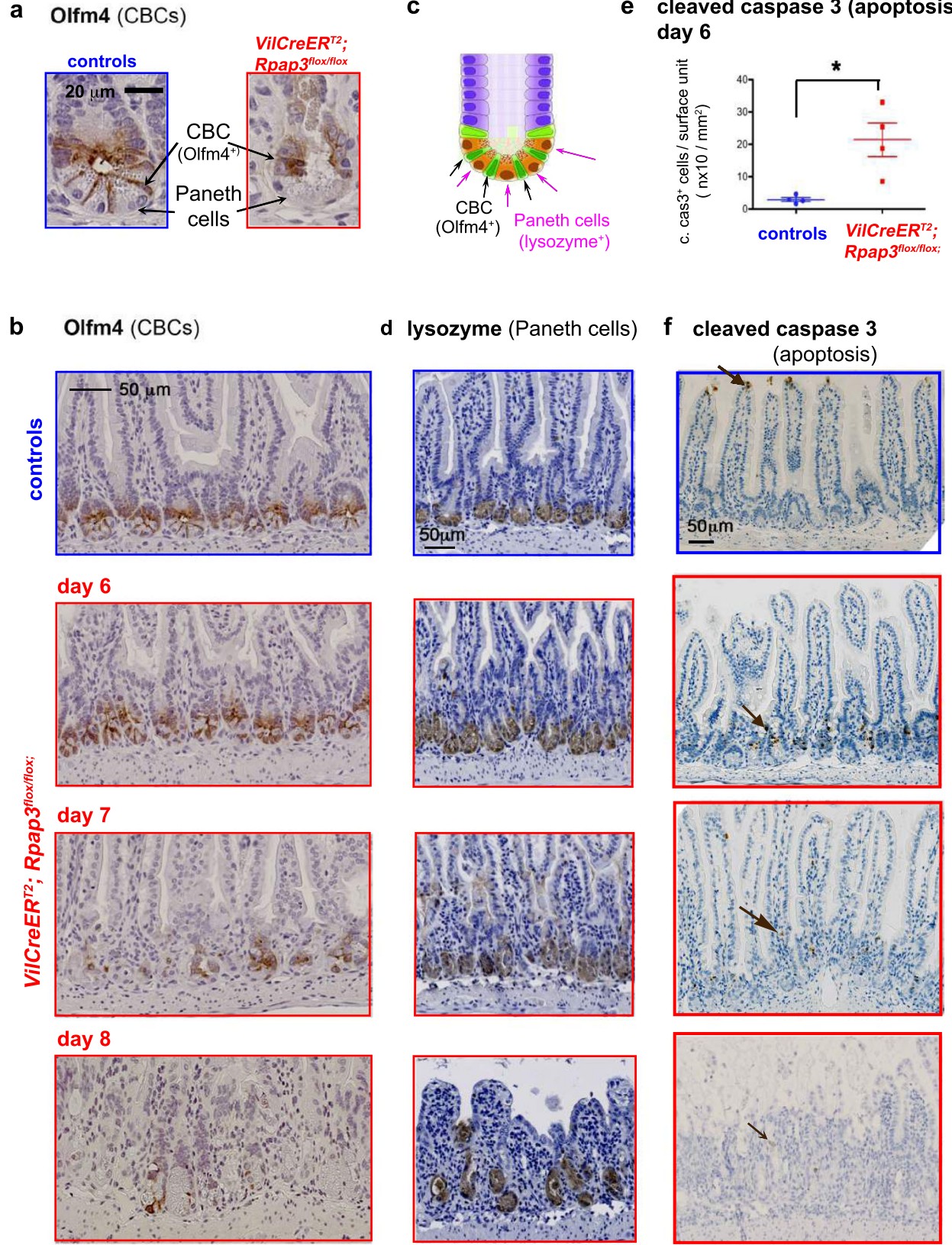

Transcriptomic analyses showed a significant enrichment of mRNAs encoding RUVBL1, RUVBL2, and RPAP3 in human primary colorectal tumors ($n = 380$) as compared to normal tissue ($n = 51$), in agreement with a previous report on a smaller cohort[50] (Fig. 9a).

This observation prompted us to test the expression of RPAP3 in CRC patient samples, using immunohistochemistry on Tissue Microarrays (TMAs) sections from CRC patients and anti-RPAP3 antibodies, which only detects human RPAP3 but not its murine homolog[51]. We analyzed TMAs containing core tissues

**Fig. 4 *Rpap3* invalidation induces CBC stem cells loss and apoptosis in the proliferative compartment of the small intestine. a, b** Staining for Olfm4 in the jejunum from control (top panel) and *VilCreER^T2; Rpap3^flox/flox* animals 7 days (**a**) or 6 to 8 days after the first tamoxifen injection. Panels are representative for 2 to 4 animals/time point from at least two independent experiments. Scale bar is 20 μm in (**a**) and 50 μm in (**b**). **c** Schematic representation of a crypt from the small intestine, with CBC stem cells (in green) sandwiched between Paneth cells (in brown) and progenitors forming the TA on top (purple). **d** Representative micrographs of tissue sections immuno-stained for lysozyme, a specific marker of Paneth cells, in the jejunum of control (top) and *VilCreER^T2; Rpap3^flox/flox* mice from day 6 to day 8. Panels are representative for 2 to 3 animals/time point from two independent experiments. Scale bar, identical in all pictures, is 50 μm. **e** Total number of apoptotic cells identified by cleaved caspase 3 (cleaved cas3+) per surface (mm$^2$) of jejunum for each mouse analyzed, at day 6. Mean values with S.E.M are indicated for each experimental group. Unpaired two-tailed *t* test with Welch's correction indicates significant difference between controls and *VilCreER^T2; Rpap3^flox/flox* animals ($p = 0.0384$; $t = 3.538$, df = 3, $n = 4$).

**f** Micrographs are tissue sections stained for cleaved caspase 3 in the jejunum. In control animals, cleaved caspase 3+ cells (brown arrows) are mainly detected at the tip of the villi, as a result of epithelial turnover (top panel). In the jejunum from *VilCreER^T2; Rpap3^flox/flox* animals, at day 6 and 7, cleaved caspase 3+ cells were detected within the crypts (brown arrows). Panels are representative from 2 to 4 animals/time point, from two independent experiments. Scale bar, identical in all pictures, is 50 μm. Source data are provided as a "Source Data file".

from patients diagnosed with CRC without pathological evidence of nodal involvement and distant metastasis. 157 out of 177 (88.7%) cases expressed RPAP3 in the tumor cell cytoplasm. The proportion of RPAP3-positive cells was in the range of 4–100%, with a mean ± S.E. of 62.6% ± 2.6. To dichotomize the RPAP3 expression level in RPAP3^high and RPAP3^low, an optimal cut-off value of 26% of positive tumor cells was chosen based on the Receiver Operating Characteristic (ROC) analysis (AUC = 0.593; Fig. 9b). The relationships between RPAP3 expression and clinico-pathological parameters were investigated by Pearson's $\chi^2$ test: RPAP3 expression negatively correlated with the tumor stage of CRC ($p = 0.049$; Supplementary Tables 1, 2). Compared with stage I tumors, the expression of RPAP3 was decreased in stage II tumors ($p = 0.049$). Furthermore, compared with mucinous carcinoma, the expression of RPAP3 was increased in CRC of the adenocarcinoma type ($p = 0.025$). In addition, high RPAP3 expression was positively correlated with the occurrence of tumor relapse ($p = 0.003$) and patients' mortality ($p = 0.036$). Since mTOR can influence CRC outcome[52], we addressed its contribution by IHC in tumors from the same patient cohort. RPAP3 contributed only to 22% of mTOR expression level (rho = 0.226, $p = 0.007$), which was not correlated with DFS (Supplementary Fig. 7).

39 out of 123 (31.7%) patients with RPAP3^high tumors and 6 out of 54 (11.1%) patients with RPAP3^low tumors had disease relapse. Analysis of Kaplan–Meier curves showed that patients with RPAP3^high tumors had a lower DFS rate than patients with RPAP3^low tumors ($p = 0.037$; Fig. 9b). Multivariate analyses of DFS adjusted for other prognostic factors confirmed that RPAP3 expression was a significant prognostic parameter influencing disease relapse (HR = 2.7: 95% CI, 1.2–6.5; $p = 0.023$; Supplementary Table 3), but not the overall survival (OS) of patients. These results provide evidence that high RPAP3-expression levels in CRC tissues are associated with poor patient prognosis.

## Discussion

Since its discovery in 2005, biochemical and structural studies highlighted a chaperoning role for R2TP[1,3]. RPAP3 is a central subunit within R2TP: it recruits the chaperones HSP70/90 via its TPR domains, PIH1D1 through a short peptide domain and the AAA+ ATPases RUVBL1/2 via its C-terminal domain[7,10–12]. Overall, R2TP appears as a central hub for assembling multisubunit complexes by coordinating the activity of HSP70/90 with that of RUVBL1/2. Most of the R2TP clients known so far have been identified in mammalian cell lines[4,15,17–20] and here we validated several of them in the intestinal epithelium. This includes PRPF8, part of the U5 spliceosomal snRNP, and NOP58, a core component of box C/D snoRNPs required for ribosomal biogenesis. Another class of R2TP substrates are the PIKKs[22]. Several studies have documented that these large proteins are

stabilized by a trimeric adaptor called TTT, which itself interacts with R2TP/HSP90[6,22,32,42,43,53]. We show here that mTOR, ATR and ATM depend on *Rpap3* in intestinal crypts. Finally, a major R2TP client is RNA PolII. This key enzymatic complex is assembled in the cytoplasm prior to its nuclear translocation[19]. The observed cytoplasmic accumulation of Rpb1 in *Rpap3* KO cells thus likely illustrates a defective assembly of RNA PolII.

Remarkably, invalidation of *Rpap3* triggers p53 stabilization, cell cycle arrest, apoptosis and destruction of the intestinal epithelium. Transcription inhibition, DNA damage and defects in ribosome biogenesis all activate p53[54–57], yet p53 induction was observed before any detectable alteration in R2TP client activity, such as DNA damage (as monitored by γH2AX; Supplementary Fig. 3a). This was even more striking in the colon, where p53 was activated while R2TP clients were only mildly affected (Fig. 7). Indeed, at day 6 and 7, p53 removal rescued the alterations in small intestine, further showing that, at this point, loss of R2TP client per se was not deleterious. To explain this result, we hypothesize that unassembled R2TP clients trigger p53 activation. For instance, exposed protein surfaces that are normally buried into complexes could sequester the p53-E3 ligase MDM2, as observed for unassembled ribosomal proteins[58]. And indeed, a subset of ribosomal proteins was recently shown to bind Rpap3[59]. These ribosomal proteins could trigger p53 activation following Rpap3 deficiency. In any case, the rescue of the *Rpap3* KO phenotype by *Trp53* removal was only transient. Defects in ATR, RNA PolII, snoRNPs, and ribosome biogenesis might induce p53-independent apoptosis and lead to small intestine degradation, as described for ATR[60] and proteins involved in housekeeping cellular mechanisms such as transcription and translation[61].

Ex vivo experiments showed that crypts from both small intestine and colon depended on *Rpap3* to form organoids, confirming the essential role of *Rpap3* in proliferating cells. Interestingly, we found that differentiated cells in the small intestine are barely affected by *Rpap3* removal, in contrast to the rapidly dividing CBC stem cells and TA progenitors. Indeed, Rpb1 accumulates in the cytoplasm of proliferating cells, while it remains fully nuclear in the differentiated cells of the epithelium. We propose that R2TP preferentially assembles the basic cellular machineries in the proliferative compartment: CBC stem cells and progenitors (Fig. 9c). Indeed, the high division rate of TA and CBC stem cells (~10–12 h) likely requires high rates of protein synthesis, folding and assembly, and accordingly, high R2TP activity[62]. In contrast, non-dividing differentiated cells may rely on the pre-existing machineries. Indeed, in Hela cells, many R2TP clients (NOP58, PRPF8, EFTUD2, ATR, and mTOR) have an half-life that is too stable to be measured (>24 h)[63]. Additional data support this model of a preferential role of R2TP in proliferating cells. First, R2TP is concentrated in CBC stem cells and progenitors of the small intestine. Second, several R2TP clients are more abundant in

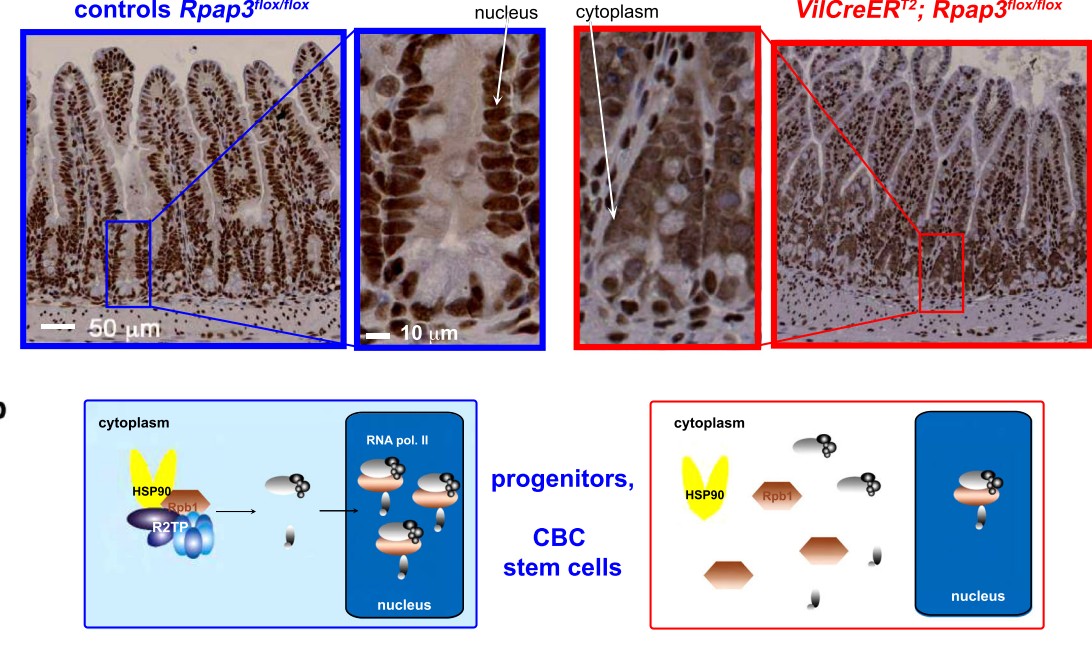

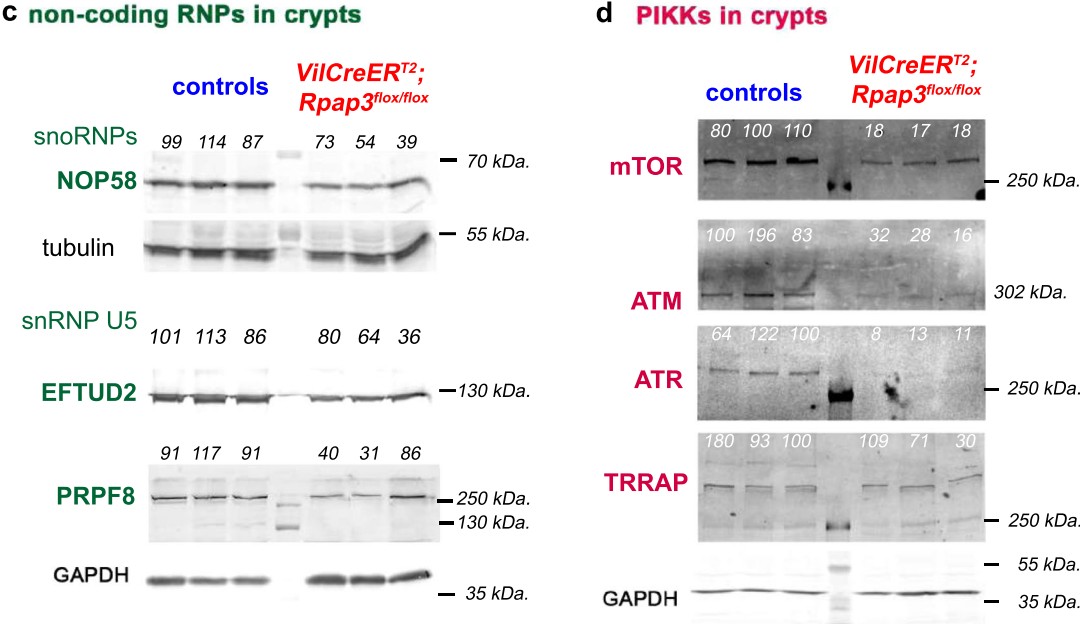

**Fig. 5 *Rpap3* deletion decreases expression of R2TP clients and leads to cytoplasmic accumulation of RNA polymerase II in intestinal crypts and TA compartment. a** Images are tissue sections of small intestines stained by immunohistochemistry (IHC) for Rpb1, the catalytic subunit of RNA polymerase II, from control *Rpap3*^flox/flox^ mice (blue frame, left panel), or *VilCreER*^T2^; *Rpap3*^flox/flox^ animals at day 6 (red frame, right panel), with magnifications of crypts (scale bars used for the two magnification insets are identical between blue and red frames). Note that the staining in stromal cells is nuclear in both wild-type and *VilCreER*^T2^; *Rpap3*^flox/flox^ animals (stromal cells do not express the Cre) and control epithelium, while it becomes cytoplasmic in the mutant epithelium. Panels are representative for *n* = 6 animals from three independent experiments. Scale bar is 50 and 10 µm for insets, as shown in control panels. **b** Schematical interpretation of the micrographs in (**a**). In control epithelial cells (blue), R2TP incorporates Rpb1 into RNA PolII, which is then imported into the nucleus. In the absence of Rpap3 (red), neo-synthesized Rpb1 accumulates in the cytoplasm. **c**, **d** Western blot analysis of preparations enriched for epithelial crypt cells from the jejunum of animals, sacrificed 6 days after the first tamoxifen injection. NOP58, EFTUD2, and PRPF8 (**c**), mTOR, ATM, ATR, and TRRAP (**d**) were detected with specific antibodies. Tubulin and GAPDH were used as loading controls. Quantification of the signal ratios are indicated on top of each lane (average for the control ratios was arbitrarily set to 100). Each lane was loaded with the lysate obtained from one animal of the indicated genotype (*n* = 3 per genotype). Similar results were obtained with animals from at least two independent experiments. Apparent molecular weights are indicated on the right. Source data are provided as a "Source Data file".

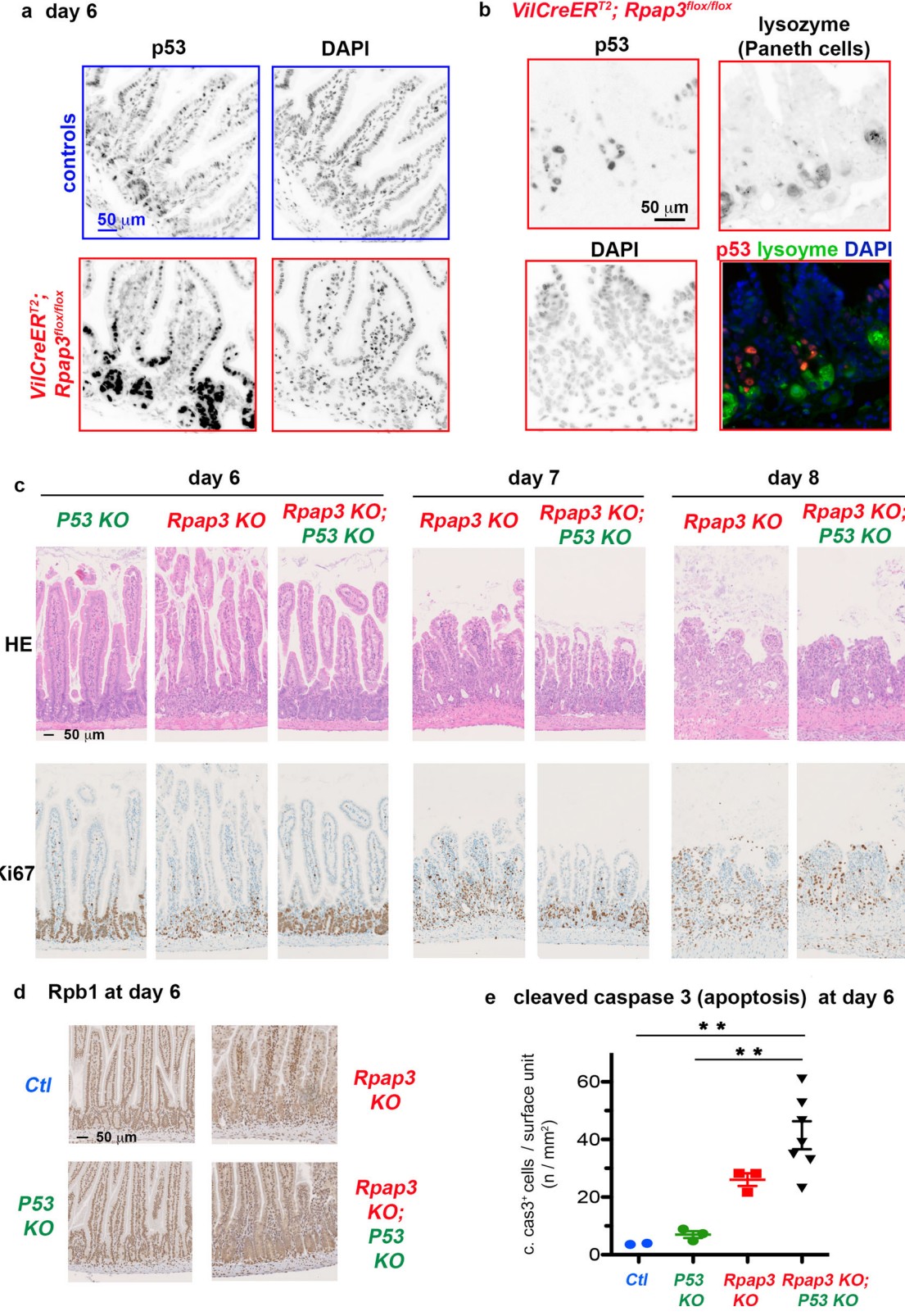

the intestinal proliferative compartment, including ATR, U3 snoRNP and U5 snRNPs (Supplementary Fig. 3b, c). Third, markers of RNA PolI activity, a potential R2TP client, accumulate in the crypts and TA compartment with decreasing levels along the crypt-villus axis[56]. Fourth, a recent transcriptomic analysis along the crypt-villus axis detected mRNAs encoding ribosomal, splicing,

and transcription components at the very bottom of the villus, but not along the villus or at the tip[64].

A correlation between proliferation and sensitivity to *Rpap3* removal further extends to the colon. Indeed, R2TP substrates showed milder destabilization in the colon than in the small intestine, while Rpb1 accumulation in the cytoplasm of

**Fig. 6 R2TP invalidation triggers p53-dependent and independent apoptosis. a** Micrographs are tissue sections stained for p53 by immunofluorescence in controls (top) and *VilCreER^{T2}; Rpap3^{flox/flox}* mice (bottom) at day 6, representative of *n = 7* animals from three independent experiments. Nuclei were stained with DAPI. Scale bar is 50 μm and is identical for all pictures. **b** Micrographs are tissue sections stained by immunofluorescence for p53 (Cy5, red) and lysozyme (Alexa 488, green), a marker of Paneth cells, in *VilCreER^{T2}; Rpap3^{flox/flox}* mice at day 6 (*n = 4*). Nuclei were stained with DAPI. Scale bar, 50 μm, is identical for all pictures. **c** Pictures of jejunum sections stained with HE (top) or by IHC with anti-Ki67 antibodies (bottom) in *P53 KO (VilCreER^{T2}; Rpap3^{flox/+}; Trp53^{flox/flox})*, *Rpap3 KO (VilCreER^{T2}; Rpap3^{flox/flox}; Trp53^{flox/+})* and double *Rpap3 KO; P53 KO (VilCreER^{T2}; Rpap3^{flox/flox}; Trp53^{flox/flox})* mice at day 6 to 8 after the first tamoxifen injection. Pictures are representative for each single KO (*n = 3*) and double KO (*n = 5–7*) mice, from three independent experiments. Scale bar, 50 μm, is identical for all pictures. **d** Pictures of jejunum sections stained by IHC with anti-Rpb1 antibody in control, *P53 KO (VilCreER^{T2}; Rpap3^{flox/+}; Trp53^{flox/flox})*, *Rpap3 KO (VilCreER^{T2}; Rpap3^{flox/flox}; Trp53^{flox/+})* and double Rpap3 KO; P53 KO (*VilCreER^{T2}; Rpap3^{flox/flox}; Trp53^{flox/flox}*) mice at day 6 following tamoxifen injection. Pictures are representative of *n = 3* for each single KO, *n = 7* for double KO, from two different experiments. Scale bar, 50 μm, is identical for all pictures. **e** Total number of apoptotic cells at day 6 per surface (mm$^2$) of the jejunum for each mouse analyzed. *n = 2* for wild type, *n = 3* for each single KO, *n = 7* for double KO, from two different experiments. Mean values with S.E.M are indicated for experimental groups with *n > 3*. One-way ANOVA analysis (*p = 0.0006*) with Bonferroni's multiple comparison post-test (**$p < 0.001$). Source data are provided as a "Source Data file".

recombinant colonic crypts was detected several days later than in the small intestine. This correlates with the rate of epithelium turnover, which is slower in the colon than in the small intestine (Fig. 8c). In Drosophila, the RPAP3 ortholog, Spag, is required for ovarian Germline Stem Cell (GSC) maintenance, but nor for the surrounding non-dividing cells of the ovary[27]. This exemplifies another case where RPAP3/Spag affects dividing stem cells, and suggests a general role of R2TP in proliferation, especially of active stem cells, across Metazoans.

Tumoral cells are particularly sensitive towards HSP90 inhibition[65–67]. This could be due to HSP90-dependent folding of proteins involved in oncogenic signaling pathways. Furthermore, an enhanced stability of the HSP90 interactome has been described in some tumors and coined the "epichaperome"[68]. In line with these, invalidation of a cytoplasmic HSP90 paralog essential for spermatogenesis[28] diminishes lung metastasis in a murine model[69]. Yet, in clinical trials, HSP90 inhibitors have been disappointing, showing cytotoxicity. One reason could be the broad range of HSP90 substrates, which also includes tumor-suppressors[70,71]. To bypass this problem Neckers and Workman suggested to target HSP90 co-chaperones that are specific for a given class of clients[72]. We propose to consider R2TP as a therapeutic target, as a role for R2TP in intestinal carcinogenesis is supported by several lines of evidences: (i) a higher expression of mRNAs encoding R2TP subunits in colorectal tumors, as compared to matching controls, (ii) high RPAP3 protein levels in biopsies of CRC patients with poor diagnostic, (iii) a strong dependency on RPAP3 in the proliferative compartment, and (iv) sensitivity of cell proliferation in a p53-dependent and independent context. R2TP is interesting not only as a therapeutic target but also for prognostic purpose, since R2TP components RPAP3 and PIH1D1 were both identified as part of the epichaperome of HSP90-sensitive tumors[68].

In conclusion, we show that the R2TP chaperone plays a crucial role in the intestinal epithelium homeostasis by promoting proliferation of stem cells and progenitors (Fig. 9c).

## Methods

**Human colorectal sample collection and analysis**. A total of 177 CRCs were collected from patients surgically treated at the University "G. D'Annunzio", Chieti, Italy, between 1996 and 2010. Only the CRC patients who did not receive adjuvant systemic therapy were included in the study. The median follow-up was 53 months (range 3–238 months). During the follow-up, 25.4% of CRC patients (45 out of 177) had a disease relapse, while deaths were observed in 18.6% of CRC patients (33 out of 177). Tumor stage was determined according to the American Joint Committee on Cancer TNM staging system (8th edition). Histologically, each CRC case was graded according to the criteria of the WHO classification of tumors of the digestive system (4th edition). Patients and tumor characteristics are summarized in Supplementary Tables 1, 2. The study was reviewed and approved by the Institutional Research Ethics Committee Comitato Etico delle Province di Chieti e Pescara e dell' Università degli studi "G. D'Annunzio" di Chieti e Pescara and written informed consent was obtained from all patients. The study complied

with all relevant regulations regarding the use of human study participants, including the criteria set by the Declaration of Helsinki.

TMAs were constructed by extracting 2-mm diameter cores of histologically confirmed neoplastic areas from the 177 CRC cases using a manual Tissue Arrayer (MTA, Beecher Instuments, WI), as previously detailed[73]. It was conducted following REMARK guidelines, as detailed in the REMARK checklist (see Supplementary Note).

**Mice generation and treatments**. Mouse experiments were performed in strict accordance with the guidelines of the European Community (86/609/EEC) and the French National Committee (87/848) for care and use of laboratory animals, comply the ARRIVE guidelines and were approved by the French Ministry of Higher Education, Research and Innovation (reference APAFIS#18685) to be performed in the institute animal facility (agreement # F3417216). Mice were housed in temperature-controlled ventilated cages (20–22 °C) with a 12 h light-dark cycle, with percentage of humidity between 45 and 55%, and maintained in pathogen-free conditions in the institute animal facility. *Rpap3^{wtsi/+}* (*Rpap3^{tm1a (KOMP)Wtsi}*) mice were generated from ES cells (JM8A1.N3, C57BL/6 N genetic background) generated by the trans-NIH KnockOut Mouse Project (KOMP) and obtained from the KOMP Repository (www.komp.org). Genotyping was performed by PCR amplification using primers F5, R5 and R6, on mouse tail genomic DNA (gDNA—Supplementary Fig. 1a). All mice strains Rpap3^{tm1a(KOMP)Wtsi}, Tg(CAG-flpo)1Afst, Tg(Vil1-cre)20Syr, Tg(Vil1-cre/ERT2)23Syr, Lgr5^{tm1cre/ERT2)Cle}, Trp53^{tm1Tyj/J} (respectively referred to as *Rpap3^{wtsi}, FlpO, VilCre, VilCreER^{T2}, Lgr5-GFP-IRES-CreER^{T2}, Trp53^{flox/flox}),RPAP3^{flox/flox}* and compounds lines were maintained on C57BL/6 background and bred in an SOPF animal facility, and during the experiment in an SPF animal facility. Naive mice were minimum 6-week-old and euthanized by CO2 and isoflurane. To activate the CreER^{T2}, controls and animals of interest received two intra-peritoneal (IP) injection of 2 mg tamoxifen each. For BrdU incorporation assays, mice were intraperitoneally injected with 100 μg Bromodeoxyuridine (BrdU) per gram of body weight. The entire small intestine (cut in three parts) and colon were flushed with PBS, then with neutral buffered formalin 10% and fixed in it for 24 h, dehydrated, and embedded in paraffin.

**Histology and immunostainings**. TMA tissue sections were stained with a mouse monoclonal antibody raised against RPAP3 (proprietary 19B11 antibody at 1:10 dilution) or an antibody against human mTOR (clone 7C10, 1:100 dilution; overnight incubation; cat. number #2983; Cell Signaling). Antigen retrieval was performed by microwave treatment at 750 W (10 min) in 10 mmol/l sodium citrate buffer (pH 6.0). The polymer kit (EnVision kit, K4003, Agilent) was used for signal amplification. DAB (3,3-Diaminobenzidine) was used as chromogen. Anti-human RPAP3 has been validated and previously published[19,51]. See Supplementary Table 4.

For histological analysis on murine tissues, tissue sections (4 μm thick) were deparaffinized and rehydrated. They were stained with hematoxylin and eosin (H&E) or Periodic Acid Schiff staining (PAS) for preliminary analysis. For Ki67, Cleaved-caspase 3 and γH2AX immunostainings, tissue sections were deparaffinized, rehydrated, and subsequently subjected to heat induced antigen retrieval in 10 mM sodium citrate buffer (pH 6) or 10 mM Tris HCl-1 mM EDTA buffer (pH 9), depending on the antibody. Immunohistochemistry was performed using a Dako autostainer (Dako, Glostrup, Denmark) by the RHEM facility.

For immunohistochemistry (Rpb1, Lysozyme, Olfm4, Pih1d1), tissue slides were incubated in 10 mM sodium citrate pH 6 (T0050, DiaPATH) or 10 mM Tris HCl 1 mM EDTA pH 9 buffer for 20 min at 100 °C for antigen retrieval, depending on the antibody. Endogenous peroxidase activity was inactivated with PBS—0.3% hydrogen (#H1009, Sigma). After blocking in 2.5% blocking serum—5% BSA—5% nonfat milk for 30 min at room temperature, tissue slides were incubated with primary antibodies overnight at 4 °C. Then corresponding secondary antibody

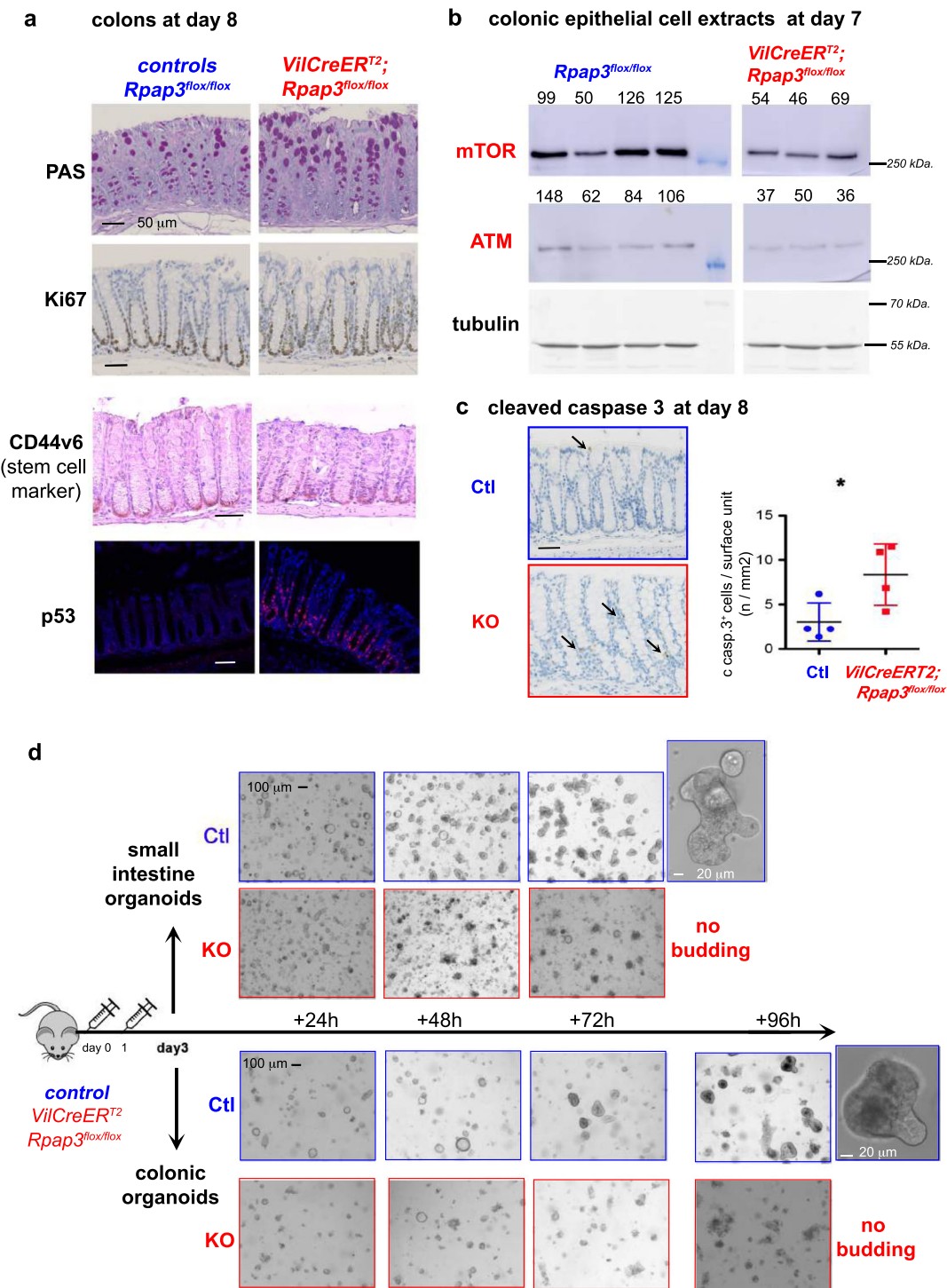

reagents (ImmPRESS™ kit, Vector Laboratories), directed against mouse, goat or rabbit were used for detection.

Incorporation of BrdU in proliferating intestinal epithelial cells was detected using an anti-BrdU antibody (Biolegend, 1:100) after deparaffinization of the tissues, antigen retrieval in 10 mM citrate buffer pH 6, as described above, and DNA denaturated using 2N HCl for 1H at 37 °C followed by an incubation in 0.1M borax buffer pH 9. Revelation was performed using the Avidin/Biotin Vectastain System kit (Vectorlab, USA) according to the manufacturer protocol. After neutralization of the endogenous peroxidase activity, the sections were incubated with the primary antibodies.

Antibody were visualized using the Envision® system (Dako). 3,3'-Diaminobenzidine (Dako) was used as the chromogen and the sections were lightly counterstained with hematoxylin. Histological slides were scanned using the

Nanozoomer 2.0 HT scanner with a ×40 objective, and visualized with the NDP. view2 software (both from Hamamatsu).

For p53 immunofluorescence, paraffin-embedded tissues were cut into 3-μm-thick sections, mounted on slides, then dried at 37 °C overnight. Staining was performed on the Discovery Ultra Automated IHC staining system from Roche Ventana. Following deparaffination with the Discovery EZ Prep solution at 75 °C for 24 min, antigen retrieval was performed at 95 °C for 16 min using the Discovery CC1 buffer. After blocking in TBS, 10% goat serum, 5% BSA, 5% milk, 0.3% triton X100, the slides were incubated with a rabbit anti-p53 antibody 37 °C for 60 min (Leica, P53-CM5P-L, 1:250). Signal enhancement was performed using the OmniMap anti-Rabbit HRP kit (Roche, 760-4457) then with the Cy5 Kit (Roche, 760-238). For double immune-stainings, slides were stripped by heating before incubation with anti-lysozyme antibody.

**Fig. 7 Rpap3 is required in the colon. a** Representative micrographs of colon sections stained by Periodic Acid Schiff stain (PAS, for labeling of goblet cells), IHC for Ki67 and CD44v6 (a marker of the colonic stem cells) and IF for p53, in controls Rpap3$^{flox/flox}$ (left) and VilCreER$^{T2}$; Rpap3$^{flox/flox}$ mice (right) at day 8 (n = 4 from at least two independent experiments). Scale bars are 50 μm. **b** Western blot analysis of colonic epithelial cells from animals, sacrificed 7 days after the first tamoxifen injection. mTOR, ATM, and tubulin were detected with specific antibodies. Quantification of the signal ratios are indicated on top of each lane (average for the control ratios was arbitrarily set to 100). Each lane was loaded with the lysate obtained from one animal of the indicated genotype (representative for n = 5 KO animals in one experiment). Molecular weights are indicated on the right. **c** Micrographs are tissue sections stained by IHC for cleaved caspase 3 in Rpap3$^{flox/flox}$ (blue) and VilCreER$^{T2}$; Rpap3$^{flox/flox}$ (red) mice at day 6, representative for n = 4 from two different experiments. Total number of apoptotic cells at day 8 per surface (mm$^2$) of the jejunum for each mouse analyzed. Mean values with S.E.M are indicated for each experimental group. Unpaired two-tailed t test with Welch's correction indicates significant difference between controls and VilCreER$^{T2}$; Rpap3$^{flox/flox}$ animals (p = 0.0234, t = 2.625, df = 5). Scale bar is 50 μm. **d** Picture of organoids cultures. Crypts from small intestine (top) and colon (bottom) were prepared from Rpap3$^{flox/flox}$ (blue) and VilCreER$^{T2}$; Rpap3$^{flox/flox}$ (red) mice 3 days after tamoxifen injection (n = 3). Identical number of crypts were seeded and organoids culture were monitored every day. Organoids from control animals started budding after 72 h in culture for small intestine crypts and 96 h for colonic crypts. Organoids generated from VilCreER$^{T2}$; Rpap3$^{flox/flox}$ animals degenerated in culture before budding. Scale bar are 100 μm for all pictures and 20 μm for organoid insets. Source data are provided as a "Source Data file".

For Rpb1/GFP immunofluorescence, antigen retrieval with 1 mM EDTA was performed for 30 min at 99 °C. Slides were blocked with 5% goat serum-PBS—0.3% triton X-100 for 1 h at RT then incubated with primary antibodies overnight at 4 °C. Samples were incubated in a DAPI solution combined with the secondary antibodies for 1 h at RT. Slides were mounted with cover slip in ProLong Gold Antifade Mounting media or Vectashield. Fluorescent images were acquired by the Axioscan 40x camera and analyzed with ZEN (both from Zeiss), brightfield upright Zeiss Axioimager Z1 using the Metamorph software, or on the inverted Confocal SP5 (Leica) using the Leica LAS AF software.

All antibodies used for immunostainings are described in Supplementary Table 4 and have been validated by the manufacturers, independent groups (see https://www.citeab.com), the RHEM platform, our previous work (Rpb1)[19] or this study.

For β-galactosidase activity, small intestines and colons cryo-preserved were cut to slices of 10 μm thickness. Samples were treated with 0.5% glutaraldehyde for 10 min at room temperature, followed by 24 h of incubation at 37 °C in a staining solution containing 1 mg/ml X-gal, 5 mM potassium ferricyanide (K3Fe(CN)6), 5 mM potassium ferrocyanide (K4Fe(CN)6) and 2 mM MgCl2, 0.1% Triton X100 in PBS. Samples were washed in PBS for 5 min, counterstained with Nuclear Fast Red and briefly rinsed with dH2O before mounting using an aqueous mounting solution (Aquatex, 108562;Merck).

**U3 snoRNA FISH.** Small intestines and colons cryo-preserved were cut to slices of 10 μm thickness. The frozen sections were fixed in PBS/paraformaldehyde 4% at room temperature for 60 min, followed by permeabilization with ethanol 70%, overnight at 4oC. In situ hybridization was performed overnight with Cy3 labeled oligonucleotides against U3 (AT*AGAACGATTCAACT*CATCAACGCGGGT*G CACTTGGCTTTCT*A).

After several washes, the slides were staining with DAPI and mounted in Vectashield.

**Microscopy and imaging.** Histological slides were scanned using the Nanozoomer 2.0 HT scanner with a ×40 objective, and visualized with the NDP.view2 software (Hamamatsu). Fluorescent images were acquired with Axioscan using ZEN software (Zeiss), brightfield Axioimager Z1 (Zeiss) using the Metamorph software or on the inverted Confocal SP5 (Leica) using the Leica LAS AF software. Images were processed with Adobe Photoshop CS6.

**Isolation of epithelial cells from the intestine.** Small intestines and colons were isolated and flushed with cold PBS. Colon and the three intestine fragments were cut open length-wise, then kept in 10 ml cold wash buffer (PBS, 2% FBS and antibiotics). Tubes were shaken several times to wash the fragments and put into 15 ml falcon tube containing 10 ml CE buffer (PBS, 1% BSA, 1 mM DTT, 1 mM EDTA, 5.6 mM glucose). Tubes were placed on a vertical shaker at 37 °C for 30 min. After removing tissue, cells were collected by centrifugation at 1410 rpm for 7 min. They were washed in 10 ml wash buffer. Collected cells were split in different tubes and stored at −80 °C.

To separate villi and crypt cells, after PBS wash, villi were gently scraped off using a glass cover slip. Free villi fractions were collected by pipetting in 1 ml of cold PBS. The rest of tissue (the crypts were still attached) was incubated with 10 ml CE lysis buffer to dissociate and collect crypt cells.

**Organoids.** Colons and small intestines were dissected, inverted on a stick, cut longitudinally then washed extensively in cold PBS-supplemented with penicillin/streptomycin. After cutting them in 0.3 cm long pieces, they were incubated in 10 ml of cold PBS. Pieces were washed 5 to 10 times in cold PBS with antibiotics and then treated for 30 min at 4 °C with PBS-25mM EDTA for the colon and PBS-2-mM EDTA for the small intestine. After sedimentation, the pieces were washed

four times in 10 ml of cold PBS. For each wash, corresponding to Fraction 1 to 4, the supernatant was strained in a 70 μm mesh filter. Each fraction was then centrifuged at 390 g for 7 min and checked for the presence of well-formed crypts under the microscope. Usually fractions 3 and 4 were used for the organoid culture. 200 crypts were resuspended in 50μl of Matrigel in WENR (DMEM-F12 media supplemented by WNt3a, EGF, Noggin, R-spondin-1) media for the colon and ENR (DMEM-F12 media supplemented by EGF, Noggin, R-spondin-1) media for the small intestine and microscopically inspected once a day, as described[47].

**Western blots.** Protein extracts were prepared from epithelial cell preparations by lysis in cold RIPA buffer (50 mM Tris pH8, 150 mM NaCl, 1% NP40, 0.5% deoxycholate) supplemented with inhibitors. Protein lysates were separated on a gradient 4–15% SDS-PAGE (BioRad) and transferred to a nitro-cellulose membrane (Amersham Protran 0.2 μm NC). Membranes were blocked with TBS —2.5% nonfat milk (w/v)—0.05% Tween and incubated with the appropriate primary antibodies, followed by an incubation with secondary antibodies conjugated to the horseradish-peroxidase or a fluorophore. ECL was revealed with Amersham ECL 600 camera and fluorescence with the Amersham Typhoon scanner.

Primary antibodies from Cell Signaling: ATM clone D2E2 (CST 2873S) rabbit at 1/1000, ATR clone E1S3S (CST 13934S) rabbit at 1/1000, mTOR (CST 2972S) rabbit 1/1000, p53 clone 1C12 (CST2524S) mouse 1/1000; PRPF8 (sc-30207, Santa Cruz) rabbit 1/200; from Abcam: EFTUD2 (ab72456) rabbit 1/2000, GAPDH (ab8245 6C5) mouse 1/10000; from Sigma: NOP58 (HPA018472) rabbit 1/100, RPAP3 (SAB1411438) rabbit 1/1000, α-Tubulin (I2G10) mouse 1/500; from Proteintech PIH1D1 (19425-1-AP) rabbit 1/1000; from Bertin TRRAP (G01043 murine clone 2D5) mouse 1/1000 (a generous gift from L. Tora). We validated anti-PRP8 in[17] and anti-TRRAP using a proprietary cell line with degradable TRRAP. All other antibodies have been validated by the manufacturers, as well as by independent groups which reported their use in several publications, notably in the CiteAb platform (https://www.citeab.com/) and are described in Supplementary Table 4.

**DNA extraction.** Samples were incubated in 100 μl of alkaline lysis buffer (NaOH 25 mM, EDTA 0.2 mM) at 92 °C for 20 min, kept on ice for 10 min. Then 100 μl of neutralizing reagent (40 mM Tris HCl pH 5) was added to the samples. 2 μl were used for each PCR.

**Southern blot.** Briefly, ES cells were incubated overnight at 65 °C in 100 mM Tris HCl pH8.8—5 mM EDTA—0.2% SDS—200 mM NaCl with 0.1 mg/ml proteinase K in a humid chamber. On the following morning, lysates were extracted twice with phenol pH7.0–8.0 and once with chloroform then precipitated in NaCl/EtOH. Following recovery, 10–15 μg of genomic DNA were digested for 24 to 48 h with appropriate highly concentrated restriction enzymes (40–50 U/μl–10 U/μg of gDNA). The analysis was performed by Southern blot as described in[74].

**PCR for genotyping.** Initial denaturation: 95 °C (2 min), Denaturation 95 °C (30 s), Annealing 55 °C (45 s), Extension 72 °C (45 s), and Final extension: 72 °C (2 min) for 35 cycles. Primers: RPAP3: F5 (GGTGCCACAGTGTGAGTG), R5 (TGCC TCCTGACTCACTACAG, R6 (ACCGTGTCGCTACGTCACTGCG); CREERT2-F (TTACGGCGCTAAGGATGACT), CREERT2-R (AAGGCCAGGCTGTTCTT CTT); P53-F (TGC TAG ATG CTT AGG GCT GC), P53-R (GAC TGG CCC TTC TTG GTC TT). Primers are reported in the Supplementary Table 5.

**Statistical analysis.** Transcriptomic analysis of R2TP subunits in human samples were extracted from the CODREAD cohort available at UCSC Xena platform for

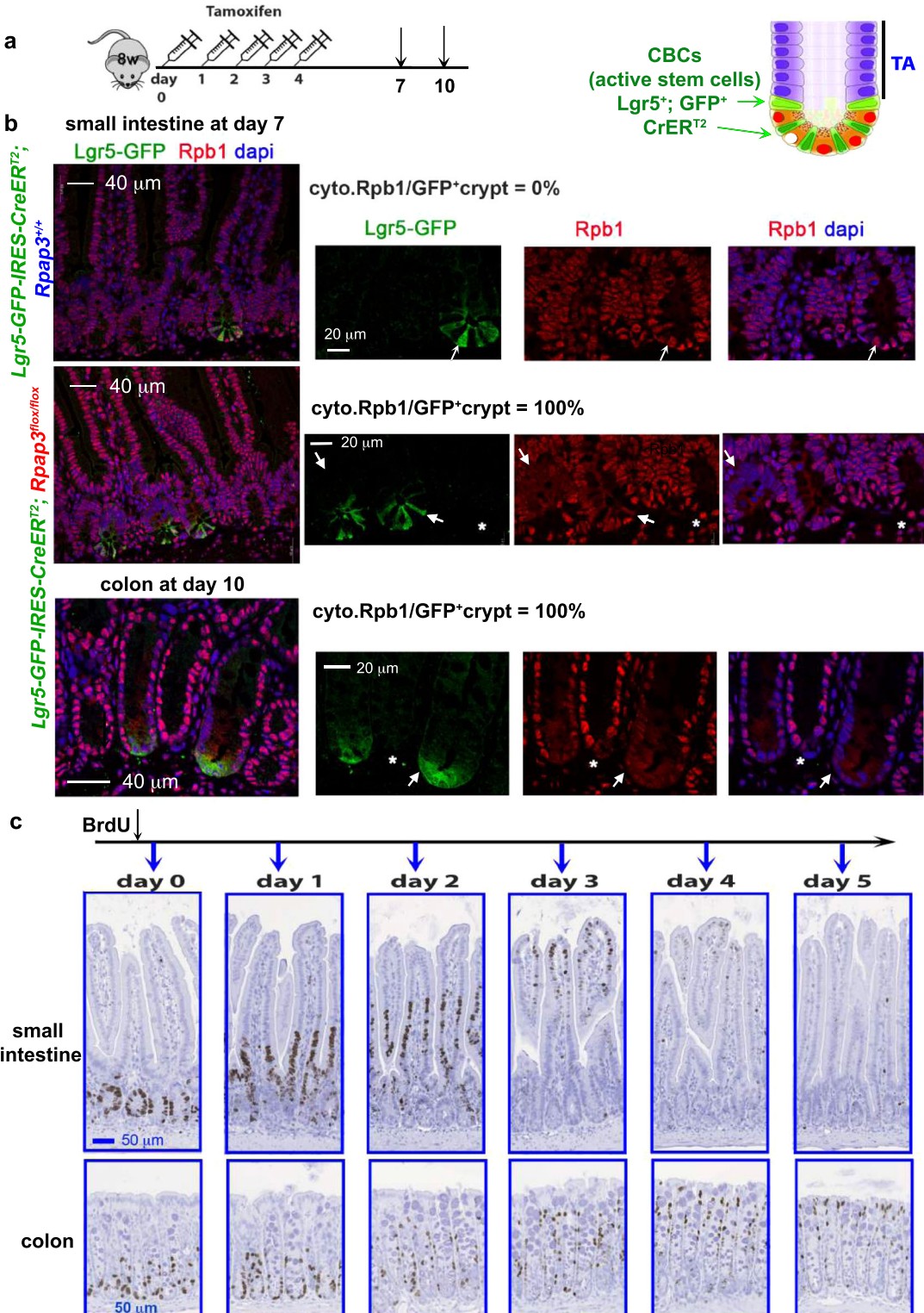

**Fig. 8 Rpap3 activity in intestinal crypts correlates with tissue turnover. a** Schematic representation of the experimental setting. Eight-week-old *Lgr5-GFP-IRES-CreER^T2^; Rpap3^flox/flox^* mice received five sequential intra-peritoneal injections of tamoxifen and were analyzed 7 or 10 days after the first injection. In this genetic model, the Cre is expressed in the Lgr5+ CBC stem cells labeled by GFP (see scheme on the right). **b** Representative images of tissue sections labeled by immunofluorescence with antibodies against GFP (green) and Rpb1 (red), with DAPI counter-staining of nuclei (blue). Please note the mosaic expression of GFP. Panels are representative for 5 animals/time point from two independent experiments. White arrow: GFP+ crypts. Asterisks: GFP− crypts. Scale bars (40 or 20 μm) are identical for matching panels. **c** Images are intestine tissue sections of wild-type animals stained for BrdU. Animals received one BrdU injection and were sacrificed at the indicated time point. The experiment was repeated twice (2 to 4 animals/time point from two different experiments). Scale bars (50 μm) are identical for all pictures.

**a R2TP mRNA levels in CR tumor tissues**

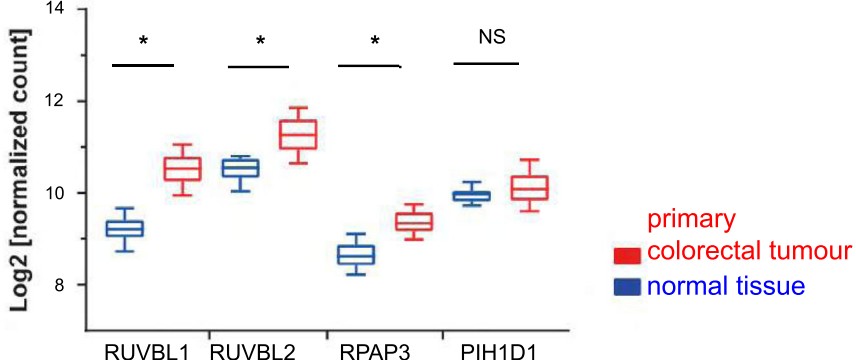

**b Disease-free survival of CRC patients according to RPAP3 in CR tumor tissues**

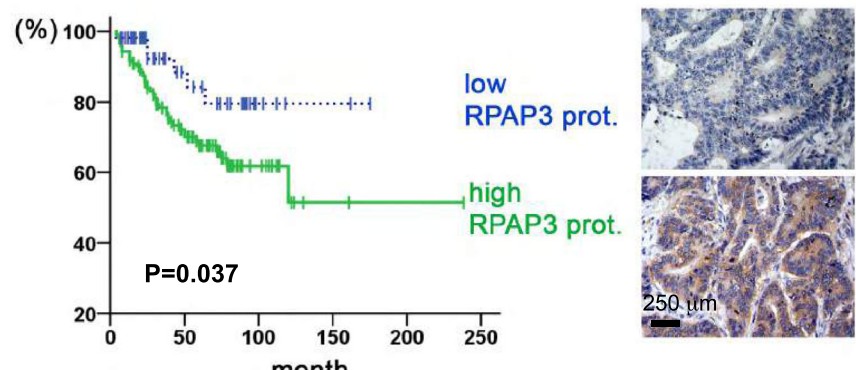

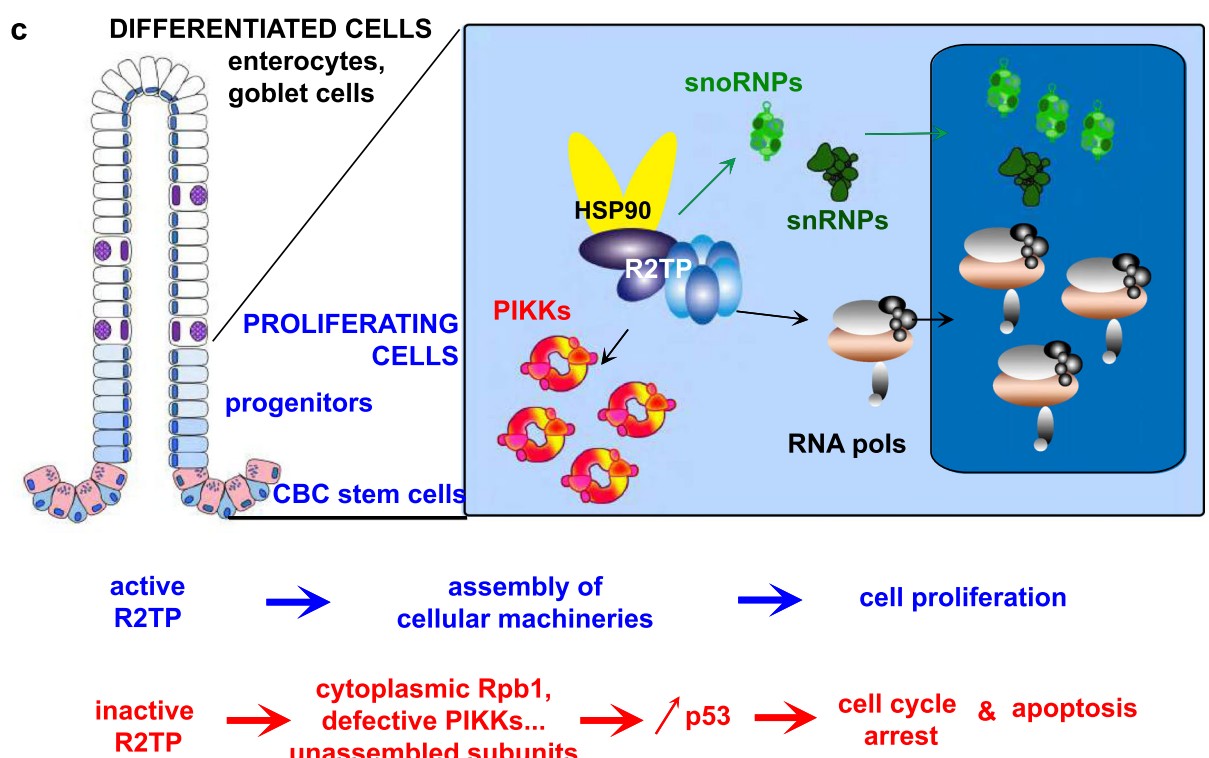

public and private cancer genomics data visualization and interpretation https://xenabrowser.net/[75]. One-way ANOVA test was performed using GraphPad Prism 5.0.

For the outcome endpoints of human data, the DFS was defined as the measure of time after treatment during which one of the following events occurred: relapse at local or distant sites, or intercurrent death without recurrence. OS was defined as the time between surgery and death from any cause. Survival curves were analyzed by the Kaplan–Meier method and compared using the log-rank test. Cox's proportional hazards model, adjusted for other prognostic factors (i.e., gender, tumor location, tumor grade, tumor stage, and RPAP3 status), was used to evaluate the association of RPAP3 expression with outcome. Spearman correlation was used to analyze the

**Fig. 9 R2TP expression correlates with pathological cell proliferation. a** The graph depicts transcript levels of R2TP components in human primary colorectal tumor samples ($n = 380$), as compared to normal solid tissues ($n = 51$) from COADREAD cohort. y-axis: Log2 normalized counts for the indicated transcript. Distributions are presented as box-and-whisker plots (center line: median; box limits, first and third quartiles; whiskers, 10th and 90th percentiles). Statistical significance was determined by one-way ANOVA (*$p < 0.001$). **b** Kaplan–Meier analysis of disease-free survival among 177 CRC patients according to the proportion of RPAP3-expressing cells in tumor tissues. Solid green line and dashed blue line indicate high and low proportion of RPAP3-expressing tumoral cells, respectively. Statistical significance was determined by log-rank test ($P = 0.037$). Right panels show examples of CRC tissues with low (top) or high (bottom) RPAP3 expression, with scale bar. **c** Proposed model for R2TP activity in the small and large intestine. R2TP assembles cellular machineries such as RNA polymerases, snoRNPs, snRNPs, and PIKKs-complexes in CBCs and progenitors in the proliferative compartment (blue cells). Differentiated cells (including Paneth cells from the small intestine crypts, in pink) mostly rely on the complexes assembled during the proliferative phase. A defect in R2TP activity induces client dysfunction, cell cycle arrest and apoptosis via p53, and eventually, epithelium degradation.

correlation between the expression of RPAP3 and mTOR. Statistical analysis were performed with the SPSS 15.0 (SPSS Inc., Chicago, IL).

All other statistical analysis were performed employing GraphPad Prism 5.0.

## Sample availability

Mouse sperm for the different strains is available upon request, as well as proprietary antibodies.

**Reporting summary**. Further information on research design is available in the Nature Research Reporting Summary linked to this article.

## Data availability

The datasets analyzed during the current study for the transcriptomic analysis of R2TP subunits in human samples (Fig. 9a) used the UCSC Xena platform for public and private cancer genomics data visualization and interpretation available at https://xenabrowser.net/[75]. The datasets concerning the patient biopsies (Fig. 9b, Supplementary Fig. 7) are not publicly available because the database contains sensitive information that could compromise patient privacy but are available. Access to the original clinical records are subject to restrictions due to legal and privacy reasons. It must be authorized by the single patient or patient's legally authorized representative, and a specific request must be issued to: Direzione Medica/Presidio Ospedaliero "SS. Annunziata"; ASL02 di Lanciano-Vasto-Chieti/Via Dei Vestini/66100 Chieti (Italy). Timeframe for response is usually 1 month. The remaining datasets that support the findings of this study are available in the Article, Supplementary Information, or, together with details about the experimental procedures, are available from the corresponding author upon reasonable request. Source data are provided with this paper.

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

## Acknowledgements

We thank Solange Morera and EMBL facility for help with the anti-RPAP3 antibody, Laszlo Tora for the gift of anti-TRRAP. The ES cells were generated by the trans-NIH KnockOut Mouse Project (KOMP) and obtained from the KOMP Repository (www.komp.org). NIH grants to Velocigene at Regeneron Inc (U01HG004085) and the CSD Consortium (U01HG004080) funded the generation of gene-targeted ES cells for 8500 genes in the KOMP Program and archived and distributed by the KOMP Repository at UC Davis and CHORI (U42RR024244). We thank the CIGM team in Institut Pasteur for microinjection experiments and animal husbandry and Laurent Le Cam for help with Trp53ᶠˡᵒˣ model. We thank the RAM, PCEA, and ZEFI animal facilities, the MRI imaging facility, and the RHEM histology facility. MRI is member of the national infrastructure France-BioImaging supported by the French National Research Agency (ANR-10-INBS-04, «Investments for the future»). RHEM is supported by SIRIC Montpellier Cancer and funded by grant INCa_Inserm_DGOS_12553, the European Regional Development foundation and the Occitanie Region. The work was supported by La Ligue Nationale Contre le Cancer (équipes labelisées to E.B., P.J.), le comité Languedoc-Roussillon de la Ligue (LS 139433-2016 to B.P. B.), and the INCa grants PLBIO 2016-161 to D.H., E.B., M.H., B.P.B., and PLBIO 2018-158 to P.J. C.M. and C.A. had fellowships from la Ligue Nationale Contre le Cancer.

## Author contributions

C.M., C.A., B.L., M.G., C.L., V.P., M.F., C.P., J.B., F.G., C. V., and B.P.B. performed experiments, F.L. generated murine model, N.T. and R.L. provided human TMA and analysis, D.H. analysed the CODREAD data, F.G., P.J., and D.H. advised on the work and commented critically on the paper. C.M., E.B., M.H., and B.P.B. conceived the study, designed the experiments, analysed the data, and wrote the paper. B.P.B. supervised the research. All authors approved the content of the paper.

## Competing interests

The authors declare no competing interest.
