## [Peer Review File · Nature Communications]

Reviewers' Comments:

Reviewer #1:

Remarks to the Author:

In this paper, Maurizy and colleagues clearly demonstrate that component of the R2TP complex Rpap3 is required for proliferation of normal stem cells and progenitors and is dispensable for post-mitotic differentiated cells in murine intestinal epithelium. The authors also show that high RPAP3 expression correlates with poor prognosis of colorectal cancer patients.

These are highly interesting results and will be of interest to scientists from variety of fields, including protein quality control, RNA biology, DNA repair and cancer biology. The work presents data from several mouse models and brings up many questions to be solved by future research (such as whether does Rpap3 play similar role in other tissues, would decreased Rpap3 expression prevent or slow down tumour growth in colon cancer mouse models, would over-expression of Rpap3 lead to increased tumourigenesis in mice, etc).

The statistical analysis seems to be appropriate and valid.

Specific comments

Page 8, line 177 – it would be good to briefly mention function of TCF4 for readers who are not from intestine homeostasis field.

Page 10, line 207 – wrongly formatted citation? (number 32 after “degraded”)

Page 11, lines 236 and 237 – comments to the Figure 4D – is the nuclear localisation of Rpb1 in the differentiated cells caused by lower levels of expression (and therefore lower dependence on the R2TP complex for its assembly) or could there be difference in overall stability of the complex in the proliferating and differentiated cells?

Figure 1D – it would be good to have a western blot for some other tissue to show that the Rpap3 levels elsewhere are not affected.

Figure 3E – the graph should indicate on the y axis that the data are from day 6.

Figure 4– for expression of Prp8 and Nop58 it would be good to get values for the signal on the western blot correlated to the tubulin or Gapdh expression to support the 2-fold reduction statement. The signal for Atr is very weak, if possible, it would be good to have a bit stronger staining (longer exposure). mTor exposure seems to be longer in the supplementary figure and it might be better to use that one in the main figure as well. Values for expression of mTor, Atr and Atm would be good, although in this case the expression very clearly decreases.

Reviewer #2:

Remarks to the Author:

This is an interesting paper presenting intriguing data on R2TP complex, which is a HSP90 co-chaperone involved in multiple cellular processes highly relevant to cancer. Recent data also showed that it can also be an important biomarker for the diagnosis and prognosis of various cancers. However, it remains unclear how this molecular chaperone complex contributes to oncogenesis. In the current work, the authors tried to elucidate the role of the R2TP complex in the intestinal homeostasis. Using a conditional knock-out mouse for Rpap3 (a central component of the R2TP complex), they claim that R2TP is critical for small intestine homeostasis. Although the proposed model is of potential biological relevance, the data presented are too preliminary. More mechanistic studies would be required to support their conclusions.

Specific Comment

1- The authors show that the RPAP3 deletion using a villin-cre mouse model did not affect the size nor the architectural organization of the colon but induced significant alterations in the small intestine. They concluded that this phenotype is caused by the putative different cellular turnover rate of cells in the small intestine as compared to those in the colon. However, they need to rule out that a potential differential expression of the components of the R2TP complex in the different areas of the intestine might underly this phenotype. The authors should compare the levels of these proteins in small intestine and colon.

2- Additionally, the authors should complete the characterization of the phenotype in the colon by

measuring apoptosis (Figure 3E), cell cycle (Figure 2C), p53 (Figure 3F) and Olfm4 levels (Figure 3B) as they did in the small intestine.

3- The authors claim that most proliferative cells are the ones that are the most sensitive to RPAP3 deletion. However, a possible alternative explanation could be the potential differential expression of RPAP3 in these cells. The authors should measure RPAP3 levels in CBC stem cells/progenitors and compared those to the expression in other intestinal cell lineages, including Paneth cells. On the other hand, they should demonstrate the actual deletion of RPAP3 in the intestinal cells in the tamoxifen-treated Lgr5-EGFP-IRES-CreERT2: Rpap3f/f mice (Figure 5).

4- In Figure 4A, the authors show that RPAP3 participated in the stabilization of ATR, ATM, mTOR, PRP8, and NOP58. Since RPAP3 deletion does not produce any phenotype (size of the organs, architectural organization, Ki67...) in the colon, a critical question would be: how are the levels of these proteins in the colon after RPAP3 deletion?

5- The authors claim that deletion of RPAP3 leads to a p53 stabilization, which induces cell cycle arrest and apoptosis in the intestinal epithelium of VilcreERT2Rpap3f/f mice. To go deeper on the mechanism, the authors should generate organoids from small intestine and colon to demonstrate the role of RPAP3 in the proliferation, complexity and viability in this ex vivo 3D culture. Does RPAP3 deletion results in high expression of p53 in these organoids? And if so, does p53 downregulation reverts cell cycle arrest and/or apoptosis in RPAP3-depleted organoids?

6- It was reported that mTORC1 activity is important for intestinal tissue homeostasis and regeneration (PMID: 29275959). Since RPAP3 deletion results in mTORC1 destabilization, the authors should clarify whether the phenotype observed in the Rpap3 knock-out intestine is due to an impairment in the mTOR activity. Is there any correlation between RPAP3 expression, mTOR expression and patient prognosis in Figure 6B?

7- The authors claim that RPAP3 deletion is associated with an increase of p53 in the crypts and the TA compartment (Figure 3F). Since RPAP3 could be dispensable in differentiated cells, how are the p53 levels in Paneth cells and enterocytes? A double staining experiments would help to solve this question.

8- Statistics are missing in Figure 1E and 1F

Answer to the reviewers

Please find our revised version of the manuscript **NCOMMS-20-01305A-Z**.

"The HSP90/R2TP assembly chaperone promotes cell proliferation in the intestinal epithelium".

We thank the reviewers for their invaluable help in improving the manuscript. We have performed and included a panel of new experiments to address the questions raised. As a result, most figures have been substantially modified (data have been incorporated into Fig.1 as Fig. 1E and Supplementary Fig. 1, 5, 9), while new ones have been created (new Fig. 6 and Supplementary Fig. 6 to address the role of p53, Fig. 7 and Supplementary Fig. 7 to present new data on colons plus organoid experiments). The text of the manuscript has been modified accordingly (most important revisions have been highlighted **in red**). We believe that this new version consolidates our previous conclusions and brings new insights, and as such, will meet your satisfaction.

On behalf of all authors,

Sincerely,

Bérengère Pradet-Balade

Specific answers to the Reviewers' comments are highlighted in blue:

Reviewer #1 (Remarks to the Author):

In this paper, Maurizy and colleagues clearly demonstrate that component of the R2TP complex Rpap3 is required for proliferation of normal stem cells and progenitors and is dispensable for post-mitotic differentiated cells in murine intestinal epithelium. The authors also show that high RPAP3 expression correlates with poor prognosis of colorectal cancer patients.

These are highly interesting results and will be of interest to scientists from variety of fields, including protein quality control, RNA biology, DNA repair and cancer biology. The work presents data from several mouse models and brings up many questions to be solved by future research (such as whether does Rpap3 play similar role in other tissues, would decreased Rpap3 expression prevent or slow down tumour growth in colon cancer mouse models, would

over-expression of Rpap3 lead to increased tumorigenesis in mice, etc).
The statistical analysis seems to be appropriate and valid.

We thank this reviewer for his positive appreciation of our work. For clarity, the referee's points have been numbered below.

Specific comments

1-Page 8, line 177 – *it would be good to briefly mention function of TCF4 for readers who are not from intestine homeostasis field.*

We have now explained that TCF4 is “the transcription factor integrating Wnt signaling, which is crucial for CBC stemness” (page 8, line 171-172).

2-Page 10, line 207 – *wrongly formatted citation? (number 32 after “degraded”)*

Thank you for carefully checking the text: the citation has been re-formatted (now reference # 348, line 205 p.10). We apologize for this error.

3-Page 11, lines 236 and 237 – *comments to the Figure 4D – is the nuclear localisation of Rpb1 in the differentiated cells caused by lower levels of expression (and therefore lower dependence on the R2TP complex for its assembly) or could there be difference in overall stability of the complex in the proliferating and differentiated cells?*

The immunohistochemistry (IHC) staining of Rpb1 indicates similar levels of expression in all cells of the epithelium (see for instance Fig. 5A). The stability of the polymerase complex in tissue is more difficult to measure, and indeed this could vary between proliferating and differentiated cells. As highlighted in the Discussion, progenitors cells need to double the amount of polymerase at each cell division, which means every ~10h in the transient amplifying compartment where cell divisions are very rapid (6 divisions take place in 48-72h, according to *Barthel, 2017* PMID: 29254493 and *Barker et al., 2008*, PMID: 18628392). This is shorter than the stability of Rpb1 measured in cell lines (~18h in proliferative HeLa cells, according to *Boisvert et al. 2011*, PMID21937730), suggesting that indeed the need to renew Rbp1 is higher in proliferating than in differentiated cells.

Another way to approach the question is to determine the location of R2TP and R2TP activity. Our experiments show that Rpap3 (as visualized by lacZ activity driven from the *Rpap3* promoter, Figure 1 B) and its partner Pih1d1 (new Fig. 1C, as this is newly generated data) are detected in crypts but not in villi. Correlatively, immune-histochemistry (IHC) or immunofluorescence (IF) analysis show a cytoplasmic accumulation of Rpb1 in the CBCs and

Transit-Amplifying (TA) compartments upon *Rpap3* invalidation (Fig. 5A). R2TP thus appears to be active in the CBCs and TA compartments from the small intestine rather than in the differentiated cells.

The higher concentration of R2TP in CBC and TA cells correlates with that of several of its substrates, reflecting a higher need for R2TP activity in these cells. In the revised version, we added data showing that other R2TP substrates, such as the snoRNP U3, (Supplementary Fig. 5B), the splicing factor PRPF8 and ATR (Supplementary Fig. 5C), have a stronger expression in crypts than in villi. This gradual distribution across the small intestinal structure coincides with that of ribosome biogenesis (Stedman et al., 2015). We propose that the assembly of R2TP substrates preferentially takes place in the proliferative compartment, and that these machineries would then be inherited by the differentiated, specialized cells where they would be very stable. We believe that the additional data we provide (Supp. Fig.5) supports this concept and make it more understandable, as explained in the discussion (line 386-400, p. 18).

4-Figure 1D – it would be good to have a western blot for some other tissue to show that the Rpap3 levels elsewhere are not affected.

Please find below a Western blot analysis that includes a panel of tissues (Rebuttal Figure 1) revealing that *Rpap3* is highly expressed in testis and barely detectable in the other tissues. This also applies to the total extracts of the small intestine, comprising epithelium and stroma (in contrast to the Western blot analysis of enriched epithelial cell preparations shown in Fig. 1 D and the rebuttal Fig. 2 below). Indeed, in the small intestinal stroma, there was no detectable activity of *Rpap3* promoter (as assayed by lacZ activity shown in Fig.1B) nor *Pih1d1* (as detected by IHC in Fig. 1C). After specific deletion of *Rpap3* in the intestine, there is no detectable diminution of *Rpap3* in testis, spleen and stomach.

In the revised version, we included PCR amplification of genomic DNA from various tissues, using a combination of three primers, to discriminate between unprocessed *Rpap3^{fllox}* and recombined *Rpap3^{Δ7}* (Supplementary Fig. 1E). This technique is more sensitive than Western blotting and the data show unambiguously that recombination of the floxed allele occurs exclusively in the small intestine and the colon, but not in the other tested tissues, as previously described (*El Marjou et al., 2004, PMID: 15282745*). Though the *Villin* promoter was described to have some activity in the kidney (*El Marjou et al., 2004, PMID: 15282745*), we did not detect any recombination, probably because of limited diffusion of intraperitoneally

injected tamoxifen to this tissue. These data are shown in Supplementary Fig.1E and reported in lines 142-143 page7.

Rebuttal Figure 1: WB analysis of *Rpap3*, *Pih1d1* and *RUVBL1* expression level in tissues from control (*Rpap3^{lox/lox}*) or knock-out (*VilCreER^{T2}; Rpap3^{lox/lox}*) that received tamoxifen injection.

5-Figure 3E – the graph should indicate on the y axis that the data are from day 6.

This has been corrected in the new Fig. 4E, we thank the reviewer for pointing this.

6-Figure 4– for expression of *Prp8* and *Nop58* it would be good to get values for the signal on the western blot correlated to the tubulin or *Gapdh* expression to support the 2-fold reduction statement.

This is now indicated on top of each lane. For each sample, we quantified the signal for the indicated protein, divided it by the signal for the control protein (GAPDH or Tubulin) and normalized the data set by arbitrarily setting the mean values in controls to 100 (now Fig. 5C, D and legends).

7-The signal for Atr is very weak, if possible, it would be good to have a bit stronger staining (longer exposure).

We replaced the picture of the first version by one with a stronger exposure (Fig. 5D in the revised version).

8- mTor exposure seems to be longer in the supplementary figure and it might be better to use that one in the main figure as well.

This has been changed as well in the new Fig. 5D.

9-Values for expression of mTor, Atr and Atm would be good, although in this case the expression very clearly decreases.

These data have been calculated as described above for PRPF8 and NOP58 (point 5 of this letter). This is indicated in the new Fig. 5D and the mode of calculation is now explained in the Figure legend.

Reviewer #2 (Remarks to the Author):

This is an interesting paper presenting intriguing data on R2TP complex, which is a HSP90 co-chaperone involved in multiple cellular processes highly relevant to cancer. Recent data also showed that it can also be an important biomarker for the diagnosis and prognosis of various cancers. However, it remains unclear how this molecular chaperone complex contributes to oncogenesis. In the current work, the authors tried to elucidate the role of the R2TP complex in the intestinal homeostasis. Using a conditional knock-out mouse for Rpap3 (a central component of the R2TP complex), they claim that R2TP is critical for small intestine homeostasis. Although the proposed model is of potential biological relevance, the data presented are too preliminary. More mechanistic studies would be required to support their conclusions.

We thank the reviewer for pointing the importance of our study. Please find below our point-by-point reply to his/her comments, which include novel experiments that detail the mechanisms involving R2TP.

Specific Comment

1- The authors show that the RPAP3 deletion using a villin-cre mouse model did not affect the size nor the architectural organization of the colon but induced significant alterations in the small intestine. They concluded that this phenotype is caused by the putative different cellular turnover rate of cells in the small intestine as compared to those in the colon. However, they need to rule out that a potential differential expression of the components of the R2TP complex in the different areas of the intestine might underly this phenotype. The authors should compare the levels of these proteins in small intestine and colon.

Please find below a comparison by WB of the R2TP component expression levels in small intestine and colon crypts, taken from 3 different animals each. This shows that the expression levels for Rpap3 and Pih1d1 are higher in small intestine crypts than in the colon (Rebuttal Fig. 2). This result is consistent with the lacZ assay (revised Fig. 1 B) and immunostaining for Pih1d1 (which are new data presented as revised Fig. 1C) - in fact, we could not detect any reliable signal for Pih1d1 in the colon. These results are explained and commented on page 6 of the revised manuscript, line 113-121. We therefore exclude the possibility that the delayed phenotype in the colon (Fig. 7, 8) is due to a slower elimination of Rpap3 – and indeed, at day 5, both tissues displayed a comparable depletion of the protein (revised Fig. 1D). On the contrary, elevated levels of R2TP in the small intestinal crypts mirrors the elevated cellular turnover in this tissue (revised Fig. 8C).

Rebuttal Figure 2: Comparison of R2TP component expression levels in between small intestinal and colonic crypts. Western blots were revealed with antibodies against the indicated proteins in extracts of the jejunum and the colonic epitheliums of $RPAP3^{fllox/fllox}$ controls animals (red). Each lane was loaded with the lysate obtained from a single animal. Signal quantification relative to GAPDH levels are indicated on top of each lane. Molecular sizes are indicated on the right.

2- Additionally, the authors should complete the characterization of the phenotype in the colon by measuring apoptosis (Figure 3E), cell cycle (Figure 2C), p53 (Figure 3F) and *Olfm4* levels (Figure 3B) as they did in the small intestine.

These data have been now included in the new Figure 7 and Supplementary Fig. 7 of the revised manuscript. Apoptosis was observed by cleaved caspase 3 (new Fig. 7C), cell cycle status by Ki67 (new Fig. 7A & Supplementary Fig. 7A) and BrdU (new Supplementary Fig. 7B), p53 by IF (new Fig. 7A and Supplementary Fig. 7D). We analysed CD44v6 as a marker for colonic stem cells (new Fig. 7A), since *Olfm4* is not expressed in the colon (Afify et al., 2016; Todaro et al., 2014). 7 days after *Rpap3* deletion, the results show no alteration for Ki67, CD44v6, BrdU (Fig. 7A & Supplementary Fig. 7A, B). However, we could detect a mild reduction in some R2TP substrates (new Fig. 7B and Supplementary Fig. 7C – see point 4 of this answer). This was accompanied by induction of p53 and cleaved caspase 3 - although less pronounced than in the small intestine. All these data are commented line 262-272 (page 12). Altogether,

these data indicate that *Rpap3* deletion affects the colon, yet with a slower kinetics than in the small intestine. This conclusion is corroborated by the organoid experiment (see hereafter the answer to point 5a: no organoids develop from either intestinal or colonic crypts that are KO for *Rpap3*) and outlined in the discussion (line 406-414, p19).

3- The authors claim that most proliferative cells are the ones that are the most sensitive to RPAP3 deletion. However, a possible alternative explanation could be the potential differential expression of RPAP3 in these cells. The authors should measure RPAP3 levels in CBC stem cells/progenitors and compared those to the expression in other intestinal cell lineages, including Paneth cells.

We tested several antibodies for detecting *Rpap3* by immunostaining but obtained no specific signal on mouse tissue sections (eventhough some antibodies work on human tissue sections). Therefore, we focused on detecting *lacZ* activity in *Rpap3^{wtsi/+}* reporter mice. This showed a clear signal in cells at the bottom of the crypts, intercalated by *lacZ* negative cells (Fig.1 B). We could not directly assess cell identity in the crypts with this method (*lacZ* activity must be performed in cryo-fixed tissues, in which cell morphology is poorly resolved), nor perform immunolabelling in parallel to the enzyme-substrate reaction (antibodies no longer worked after the *lacZ* labelling reaction). Yet, we were able to identify an antibody directed against *Pih1d1* that works in IHC. *Pih1d1* staining of the small intestine clearly shows a signal in CBCs and in the TA, but not in Paneth cells nor villi. This result has been now included in the new Figure 1C of the revised manuscript. Since *Pih1d1* is an obligated partner for *Rpap3* and is unstable without it (Fig. 1A; Izumi et al., 2012, PMID: 21951644), these data show that R2TP is present in CBCs and TA cells, but not in Paneth cells nor enterocytes. We believe that these data definitely exclude the possibility that high R2TP expression levels in enterocytes and Paneth cells account for their lack of phenotype after *Rpap3* KO. These results are commented on lines 113-121 (page 6).

On the other hand, they should demonstrate the actual deletion of RPAP3 in the intestinal cells in the tamoxifen-treated Lgr5-EGFP-IRES-CreERT2: Rpap3^{f/f} mice (Figure 5).

We performed RT-PCR analysis on GFP⁺ cells isolated from small intestines of *Lgr5-EGFP-IRES-CreERT2; Rpap3^{lox/lox}* (Rebuttal Figure 3). In these cells, we detected amplicons specific to *Rpap3^{Δ7}*, but not in GFP⁺ cells from control animals (*Lgr5-EGFP-IRES-CreERT2; Rpap3^{+/+}*). This demonstrates an effective recombination in *Lgr5⁺* CBCs from KO mice. Accordingly, RT-qPCR assays show a drastic diminution of full length *Rpap3* mRNAs

in samples prepared from GFP⁺ cells of *Lgr5-EGFP-IRES-CreERT2;Rpap3^{fllox/fllox}* animals treated with tamoxifen (*Rpap3^{wt}* mRNAs/ total *Rpap3* mRNAs).

RT-PCR on sorted GFP+ cells

Rebuttal Figure 3: RT-PCR on sorted GFP⁺ intestinal epithelial cells from *Lgr5-GFP-IRES-CreERT2; RPAP3^{+/+}* (controls, lanes #1, 2) and *Lgr5-GFP-IRES-CreERT2; RPAP3^{fllox/fllox}* (lanes #3, 4) mice. (left) Amplicons specific for *Rpap3^{Δ7}* mRNA are amplified in cDNAs from *Lgr5-GFP-IRES-CreERT2; RPAP3^{fllox/fllox}* samples but not from *Lgr5-GFP-IRES-CreERT2; RPAP3^{+/+}* controls. Molecular marker is indicated on the left. (right) Schematic representation of primer pairs discriminating between full length *Rpap3* cDNAs (“*Rpap3⁺*”) or both full length and recombined *Rpap3^{Δ7}* (“total”). RT-qPCR ratios of amplicons corresponding to full length *Rpap3^{wt}* mRNA on all *Rpap3* mRNAs (full length or $\Delta 7$) in cDNAs from *Lgr5-GFP-IRES-CreERT2; RPAP3^{+/+}* controls and *Lgr5-GFP-IRES-CreERT2; RPAP3^{fllox/fllox}* animals.

4- In Figure 4A, the authors show that RPAP3 participated in the stabilization of ATR, ATM, mTOR, PRP8, and NOP58. Since RPAP3 deletion does not produce any phenotype (size of the organs, architectural organization, Ki67...) in the colon, a critical question would be: how are the levels of these proteins in the colon after RPAP3 deletion?

We answered this question by comparing expression levels in epithelial cells from colon and small intestine using Western blot. mTOR, ATM and PRP8 expression levels diminished upon *Rpap3* depletion, but to a lower extent than in the small intestine (mTOR, ATM diminished about 2-fold in the colon and 5-fold in the small intestine; see new Figure 7B and Supplementary Fig. 7C, lines 265-268 p.12). In addition, we did not detect Rpb1 accumulation in the cytoplasm of colonic epithelial cells by IHC (Supplementary Fig. 7D; see comment lines 268-270 page 12). This is consistent with the results obtained with the mosaic *Lgr5-GFP-*

CreER^{T2}; Rpap3^{lox/lox} model, in which Rpb1 cytoplasmic accumulation in the colon is detected at day 10 (a point that most *VilCreER^{T2}; Rpap3^{lox/lox}* mice could not reach), but not at day 7 (Fig. 8B and Supplementary Fig. 8).

5a-The authors claim that deletion of Rpap3 leads to a p53 stabilization, which induces cell cycle arrest and apoptosis in the intestinal epithelium of VilcreERT2Rpap3ff mice. To go deeper on the mechanism, the authors should generate organoids from small intestine and colon to demonstrate the role of Rpap3 in the proliferation, complexity and viability in this ex vivo 3D culture.

We appreciated the value of this suggestion and generated organoids from small intestine and colon, of both control and *VilcreERT2; Rpap3^{lox/lox}* mice that had previously received tamoxifen injection. We derived the organoids from both tissues simultaneously from the same individuals (new Fig. 7D). First, we observed that in control mice, budding of colonic organoids followed that of the small intestine delayed by about 24hrs, which again supports a slower tissue turnover in the colon as compared to the small intestine. Second, organoids derived from KO mice never achieved budding, showing that *Rpap3* is necessary for CBC and progenitor survival and differentiation. Importantly, this was observed for crypts from small intestine and colon (again with a 24 hrs delay), demonstrating that *Rpap3* affects colonic crypt biology, with a delay respective to the small intestine. These important results confirm our *in vivo* data and provide an extremely solid demonstration of the role of R2TP in proliferating cells. The organoid results are presented in the new Figure 7D and commented on lines 273-282 of the revised manuscript (page 12-13) and 386-387 (page 18).

5b-Does RPAP3 deletion results in high expression of p53 in these organoids? And if so, does p53 downregulation reverts cell cycle arrest and/or apoptosis in RPAP3-depleted organoids?

We sought to answer this question by generating double KO *VilcreER^{T2}; Rpap3^{lox/lox}; Trp53^{lox/lox}* mice, to allow a careful characterization of the organs. Single *p53* KO mice display no overt phenotype, with no alteration of the intestine during the short time-course of the experiments, as described by others (see for instance *Chanrion et al., 2014*, PMID: 25295490; *Schwitalla et al., 2013*, PMID: 23273920). Interestingly, *p53* depletion rescued the phenotype observed upon *Rpap3* depletion at day 6 (including epithelium integrity and Ki67 staining) but not at day 8 (see new Fig. 6 C-E and Supplementary Fig. 6 C, D). Rpb1 accumulation in the cytoplasm of epithelial cells was identical in single *Rpap3* KO and double *Rpap3 p53* KO,

demonstrating similar penetrance and kinetics and immediate consequences in R2TP loss-of-activity. These results are documented in a new section on pages 11 and 12 (line 239-258).

Altogether, our data suggest that defective R2TP triggers p53 and in turn, cell cycle arrest and apoptosis, before there is a substantial lack of R2TP client activity. Unassembled subunits could be responsible for triggering p53 activation, as described for misassembled or defective ribosomal proteins (*Zhang et al., 2006* PMID: 16893887; *Nicolas et al. 2016* PMID 27265389). At day 8, defective activity in ATR, RNA PolIII, snoRNPs could be responsible for apoptosis and tissue degradation, independently of p53. Indeed, in an unbiased screen in colonic cells, depletion of proteins involved in transcription and protein synthesis induced p53-independent apoptosis (*Krastev et al., 2011*, PMID: 21642980), and this has also been described for *ATR* inactivation (*Ruzankina et al., 2009* PMID: 19718024)- see below). These new results illustrate the co-existence of two separate pathways that ultimately result in apoptosis and epithelium degradation, as discussed in the new version of the manuscript on pages 17 and 18 (line 369-385).

6a- It was reported that mTORC1 activity is important for intestinal tissue homeostasis and regeneration (PMID: 29275959). Since RPAP3 deletion results in mTORC1 destabilization, the authors should clarify whether the phenotype observed in the Rpap3 knock-out intestine is due to an impairment in the mTOR activity.

As reported by Brandt. & col. (PMID: 29275959), Sampson et al. (PMID: 26631481) and Barron et al. (PMID: 28275690), mTOR activity is necessary for the development of intestinal epithelium, and regeneration following irradiation or resection, but not for its homeostasis – at least not during the short one-week time-course of our experiments (in PMID: 29275959 and PMID: 28275690, phenotypes were observed 2 weeks after manipulation of mTOR activity). It is thus unlikely that the observed decrease in mTOR would be responsible for the rapid degeneration of the epithelium that we observed.

ATM and DNA-PKcs are also dispensable for intestinal epithelium maintenance (*Barlow et al., 1996* PMID: 8689683) (*Gurley and Kemp, 2007*, PMID: 18171989). In contrast, *ATR* deletion compromises intestinal epithelium integrity within 7-12 days post-recombination, inducing a phenotype resembling *Rpap3 KO* mice (*Ruzankina et al., 2007*, PMID: 18371340). Yet, in our model, unlike in *ATR KO* mice, we did not detect an increase of γ H2AX, a hallmark of DNA damage (this piece of data has been integrated in Supplementary Fig. 5A). Of note, deletion of *Trp53* exacerbated the effect of *ATR* deletion, while it transiently

rescued *Rpap3* deletion (Ruzankina et al., 2009, PMID: 19718024). Altogether, these data suggest that a yet unknown mechanism induces p53-dependent apoptosis early upon R2TP invalidation, while deficient *ATR* might participate to the epithelium degradation at later time-point. This point is now more clearly explained in the discussion (lines 382-385 on page 18).

6b-Is there any correlation between RPAP3 expression, mTOR expression and patient prognosis in Figure 6B?

We re-analysed the patient data after performing mTOR IHC in patient tissues. Correlation analysis estimates that *Rpap3* contributes for 20% of mTOR expression levels in tissue samples. Besides, mTOR levels were not significantly correlated with patient prognosis. These data are now presented in Supplementary Table 1-3, Supplementary Fig.9 and commented on page 15 (line 341-343). Few conflicting reports about mTOR level and CRC patient outcome exist. In fact, while activation of the Akt-mTOR pathway is a frequent event in CRC, this activation is generally due to upstream mutations rather than in increased mTOR level itself (Francipane and Lagasse, 2014, PMID: 24393708 ; Muzny et al., 2012, PMID: 22810696). We believe that high RPAP3 levels contribution to CRC outcome involves several of its clients, rather than mTOR alone.

7- The authors claim that RPAP3 deletion is associated with an increase of p53 in the crypts and the TA compartment (Figure 3F). Since RPAP3 could be dispensable in differentiated cells, how are the p53 levels in Paneth cells and enterocytes? A double staining experiments would help to solve this question.

Staining of p53 is restricted to the crypts from the small intestine (Fig. 6A), and we provide new data showing that it is also the case in the colon (new Figure 7A and Supplementary Figure 7E). To check p53 expression in Paneth cells, we performed double-immunostainings for p53 and lysozyme, a marker specific for Paneth cells. This showed mutually exclusive signals, unambiguously ascribing p53 expression in CBCs and progenitors (Figure 6B and Supplementary Fig. 6B, commented on page 11, lines 234-238).

8- Statistics are missing in Figure 1E and 1F

We performed appropriate statistic tests and indicated the results in Figure 1E and 1F and the corresponding legends. We thank the reviewer for pointing this omission.

Reviewers' Comments:

Reviewer #1:

Remarks to the Author:

The authors have addressed all my comments and questions and they have added a substantial amount of data which corroborate their previous finding. I recommend the paper for publication.

Reviewer #2:

Remarks to the Author:

The authors have satisfactorily addressed my concerns.

Point-by-point answer to the reviewers.

REVIEWERS' COMMENTS

Reviewer #1 (Remarks to the Author):

The authors have addressed all my comments and questions and they have added a substantial amount of data which corroborate their previous finding. I recommend the paper for publication.

Thank you.

Reviewer #2 (Remarks to the Author):

The authors have satisfactorily addressed my concerns.

Thank you.